# Intrinsic cooperativity potentiates parallel *cis*-regulatory evolution

Trevor R Sorrells[1,2]*, Amanda N Johnson[3], Conor J Howard[1,2],
Candace S Britton[1,2], Kyle R Fowler[1,2], Jordan T Feigerle[3], P Anthony Weil[3],
Alexander D Johnson[1,2]*

[1]Department of Biochemistry and Biophysics, Tetrad Graduate Program, University of California, San Francisco, United States; [2]Department of Microbiology and Immunology, University of California, San Francisco, United States; [3]Department of Molecular Physiology and Biophysics, Vanderbilt University School of Medicine, Nashville, Tennessee

**Abstract** Convergent evolutionary events in independent lineages provide an opportunity to understand why evolution favors certain outcomes over others. We studied such a case where a large set of genes—those coding for the ribosomal proteins—gained *cis*-regulatory sequences for a particular transcription regulator (Mcm1) in independent fungal lineages. We present evidence that these gains occurred because Mcm1 shares a mechanism of transcriptional activation with an ancestral regulator of the ribosomal protein genes, Rap1. Specifically, we show that Mcm1 and Rap1 have the inherent ability to cooperatively activate transcription through contacts with the general transcription factor TFIID. Because the two regulatory proteins share a common interaction partner, the presence of one ancestral *cis*-regulatory sequence can 'channel' random mutations into functional sites for the second regulator. At a genomic scale, this type of intrinsic cooperativity can account for a pattern of parallel evolution involving the fixation of hundreds of substitutions.
DOI: https://doi.org/10.7554/eLife.37563.001

## Introduction

Contingency is widespread in evolution, with chance historical changes dictating the repertoire of future possibilities (*Bershtein et al., 2006*; *Bloom et al., 2006*; *Blount et al., 2012*; *Ortlund et al., 2007*; *Sorrells et al., 2015*; *Starr et al., 2017*). Nevertheless, repeated evolutionary events in different lineages demonstrate that evolution is, at least in some instances, predictable. Repeatability is broadly referred to as convergent evolution, and when a trait arises repeatedly by a similar molecular mechanism, it is referred to as parallel evolution (*Stern, 2013*). Beginning with Darwin (*Darwin, 1883*), convergence has been taken as evidence for adaptation, but it can also be caused by drift within the constraints that arise from the properties of biological systems (*Losos, 2011*). Instances of repeated events can be thought of as natural experimental replicates for finding general principles that result in convergent evolution, and this information could be used to predict which evolutionary paths are most probable.

Genomic studies have found sets of genes that underlie convergent evolution of a wide variety of traits from across the tree of life (*Bellott et al., 2010*; *Denoeud et al., 2014*; *Gallant et al., 2014*; *Mycorrhizal Genomics Initiative Consortium et al., 2015*; *Marvig et al., 2015*; *McCutcheon et al., 2009*; *Nagy et al., 2014*; *Pfenning et al., 2014*; *Sherwood et al., 2014*; *Soria-Carrasco et al., 2014*). Such sets of genes tend to evolve in functionally related groups and are controlled by *cis*-regulatory sequences for particular transcription regulators, forming a transcription network. One way that an entire network can be up- or down-regulated is through alterations in the expression of a 'master' regulator, propagating a new expression level to all of its downstream target genes

*For correspondence:
trevorsorrells@gmail.com (TRS);
ajohnson@cgl.ucsf.edu (ADJ)

Competing interests: The authors declare that no competing interests exist.

**eLife digest** Sometimes evolution repeats itself. For example, independent butterfly species can evolve the same warning pattern to ward off predators. In many cases, the reason that a certain trait crops up again and again in parallel evolution is unknown.

One example is from the evolution of fungi, where a particular DNA sequence appeared several times independently in a large range of genes in different fungus species. This DNA sequence binds to a protein called Mcm1, which regulates nearby genes. Exactly why this DNA sequence has evolved in parallel so often in fungi has not been clear until now.

Researchers want to find out what is so special about this DNA binding sequence for Mcm1, as there are many other proteins that could do the same job. Now, Sorrells et al. investigated this further by testing whether a binding site for another protein Rap1 often found close by had a role to play. Experiments using 162 different fungus species showed that Mcm1 binding sites had evolved 12 times in parallel. Rap1 and Mcm1 did indeed turn out to work together to regulate nearby genes. The two proteins interact with a large protein complex critical for activating genes.

As a result, Mcm1 binding sites are more likely to evolve and play a role in gene regulation in different species when they are located near Rap1 binding sites. This could explain why this particular DNA sequence has evolved in parallel so many times. The same principle may apply to other genetic sequences involved in parallel evolution. With this understanding, it could be possible to predict when and where this event might occur in the future in fungi. This could be particularly useful for working towards being able to predict and anticipate the evolution of drug-resistant fungal pathogens.

DOI: https://doi.org/10.7554/eLife.37563.002

(*Chan et al., 2010*; *McGregor et al., 2007*; *Rebeiz et al., 2009*; *Reed et al., 2011*). However, in many cases—instead of changing the expression pattern of a master regulator—each gene in the network independently acquires the same *cis*-regulatory sequence, requiring hundreds of mutations across the genome (*Booth et al., 2010*; *Borneman et al., 2007*; *Gasch et al., 2004*; *Kasowski et al., 2010*; *Lowe et al., 2011*; *Paris et al., 2013*; *Piasecki et al., 2010*; *Schmidt et al., 2010*; *Tanay et al., 2005*; *Tuch et al., 2008b*). An outstanding question is how this process occurs.

Several molecular mechanisms for gene-by-gene rewiring have been proposed (*Britten and Davidson, 1971*; *Tuch et al., 2008b*). Hitchhiking of a *cis*-regulatory site on a transposable element is a mechanism well-documented in plant and animal evolution, and can rapidly bring genes under new regulatory control (*Bourque et al., 2008*; *Chuong et al., 2016*; *Kunarso et al., 2010*; *Lynch et al., 2011*; *Schmidt et al., 2012*). Alternatively, one transcription regulator can gain a protein-protein interaction with another, followed by the evolution of *cis*-regulatory sequences (*Baker et al., 2012*; *Lynch et al., 2008*; *Pérez et al., 2014*; *Tsong et al., 2006*). This latter mechanism is able, at least in principle, to rewire an entire set of genes at once upon evolution of the new protein-protein interaction; the individual gains of binding sites could occur secondarily. Nevertheless, many—if not most—examples of network rewiring seem to have occurred in the absence of evidence for either of these two mechanisms (*Borneman et al., 2007*; *Gasch et al., 2004*; *Kasowski et al., 2010*; *Lowe et al., 2011*; *Paris et al., 2013*; *Piasecki et al., 2010*; *Schmidt et al., 2010*; *Tanay et al., 2005*; *Tuch et al., 2008a*).

Here, we studied an example of transcription network rewiring in which ~100 ribosomal protein genes gained binding sites for the Mcm1 transcription regulator, and we describe evidence supporting a new mechanism for the concerted gains. The gain of Mcm1 binding sites in the ribosomal protein genes occurred in parallel in two respects: (1) the *cis*-regulatory sites were gained upstream of a large proportion of the ribosomal proteins in each species; and (2) they were gained independently approximately 12 – 13 times during Ascomycete evolution. At each gene, several point mutations were probably necessary to produce a close match to the 16-basepair Mcm1 binding site, thus requiring several hundred mutations across the entire gene set.

Based on results from a variety of experimental approaches, we argue that the gain of Mcm1 sites was potentiated in several different clades by the presence of an ancestral transcription regulator of the ribosomal protein genes, Rap1. We demonstrate that Rap1 and Mcm1 have the intrinsic ability

to cooperate in the activation of transcription, even when artificially introduced in species in which their sites are not found together. Biochemical and genetic experiments show that both regulators interact with the general transcription factor TFIID. We propose that the intrinsic, ancient ability of both proteins to interact with a common component of the general transcription machinery facilitated the repeated evolution of the Mcm1 sites in independent lineages.

## Results

### Gains of functional Mcm1 *cis*-regulatory sites

Previously, genome-wide chromatin immunoprecipitation and bioinformatics experiments revealed that Mcm1 *cis*-regulatory sites evolved independently at the ribosomal protein genes (RPGs) in several yeast lineages (*Tuch et al., 2008a*). To expand this analysis, we used 162 sequenced fungal genomes, identified RPG regulatory regions, and searched for *cis*-regulatory sequences for Mcm1 and 11 other transcription regulators that are known to regulate ribosomal components in at least one species (*Figure 1A*, *Figure 1—figure supplement 1*). Mcm1 sites were found highly enriched (-log10($P$) > 6) at the RPGs in six different monophyletic groups including the clades represented by *Kluyveromyces lactis*, *Candida glabrata*, and *Yarrowia lipolytica*, as well as the individual species *Kazachstania naganishii*, *Pachysolen tannophilus*, and *Arthrobotrys oligospora*. Mcm1 sites were found moderately enriched (6 > -log10($P$) > 3) in the RPGs in seven additional lineages.

The pattern of yeast lineages containing Mcm1 binding sites upstream of the RPGs could occur through three general scenarios: the Mcm1 binding sites were gained multiple times, they were present in an ancient ancestor and lost multiple times, or a combination of gains and losses. To estimate how many gains and losses occurred during the evolution of Mcm1 sites, we treated the presence of Mcm1 sites (-log10($P$) > 3) as a discrete character and sampled 10,000 stochastic character maps from the posterior probability distribution of each of two models (*Bollback, 2006*; *Revell, 2012*) (*Figure 1B*). One model assumed gains and losses to be equal in probability and the other estimated them as two different rates. Under both models, the number of gains was substantial: an average of 13.4 and 12.0 for the equal and different rates models, respectively (95% highest posterior density interval of 12 – 14 and 7 – 17 gains, respectively). The number of losses differed between the models resulting in an average of 3.0 losses for the equal rates model and 17.0 for the different rates model. These models indicate that the number of times Mcm1 sites were gained in the Ascomycete evolution was high (12 – 13 on average) whereas the number of losses is sensitive to the assumptions of the model.

Three observations support the idea that the pattern of Mcm1 *cis*-regulatory sequences in the RPGs require independent gains for most of the monophyletic clades and species with Mcm1 sites. First, the gain model is more parsimonious because of the sparse distribution of clades with Mcm1 sites. This is still true when taking into account the ancient interspecies hybridization that occurred in two ancient members of *Saccharomycetaceae* (*Marcet-Houben and Gabaldón, 2015*) because in most likely scenarios, the ancestors are concordant for the absence of Mcm1 sites. Second, other *cis*-regulatory sequences show similar patterns of evolution to that of Mcm1. The Dot6/Tod6 and Rim101 motifs are also found upstream of the RPGs in several distantly related clades, although fewer clades than for Mcm1 (*Figure 1—figure supplement 1*). Other regulators show clear single gains in the common ancestor of all Saccharomycotina yeasts (e.g. Rap1), or Pezizomycotina and Saccharomycotina (e.g. Cbf1) as well as losses in sparsely distributed individual clades and species. These results show that gains and losses of *cis*-regulatory sequences are common upstream of the RPGs, and that they can be distinguished from each other based on their distinct distributions among species (*Lavoie et al., 2010*; *Tanay et al., 2005*). Third, it was previously shown that the entire set of genes that Mcm1 regulates changes extensively over the timescale of Ascomycete evolution (*Tuch et al., 2008a*), suggesting that conservation from a distant ancestor would be a marked exception to this trend.

To test whether the Mcm1 *cis*-regulatory sites we identified upstream of the RPGs are functional, we linked several full-length RPG upstream intergenic regions to the fluorescent reporter GFP. We chose *RPL37* and *RPS18* from *Kl. lactis*, each of which has Mcm1 binding sites and measured the expression of these reporters under nutrient-replete conditions. To test whether they contributed to expression, we scrambled the Mcm1 sites. In both cases, the scrambled site reduced expression of

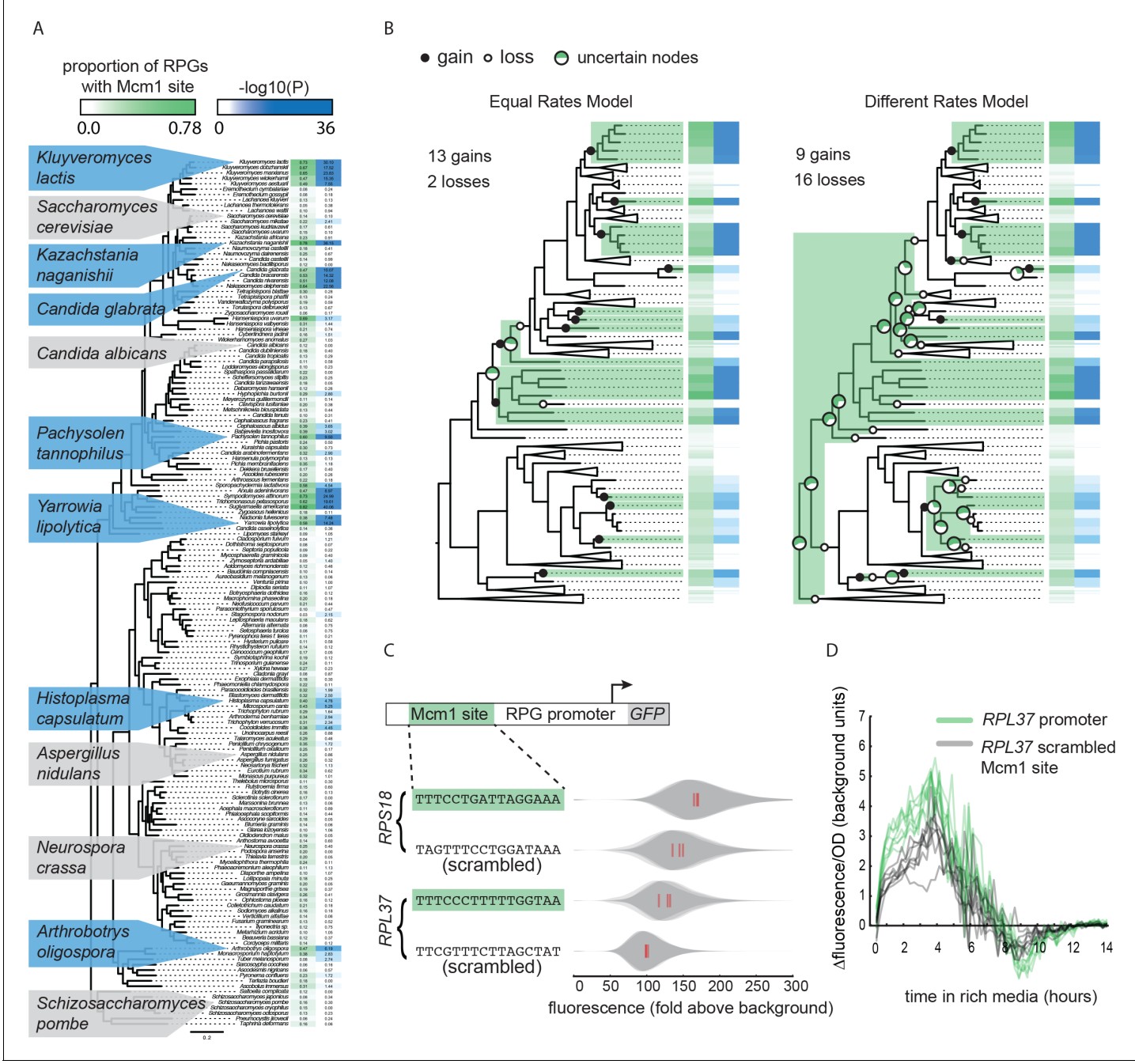

**Figure 1.** Repeated evolution of Mcm1 *cis*-regulatory sites at RPGs. (**A**) The Mcm1 sites are found upstream of the ribosomal protein genes (RPGs) in several different clades in the Ascomycete fungi. The first column, colored in green, shows the proportion of ribosomal proteins in each species that contains at least one Mcm1 site at a cutoff of ~50% the maximum log likelihood score using a position weight matrix. The second column, colored in blue, shows the –log10(*P*) for the enrichment of these Mcm1 binding sites relative to upstream regulatory regions genome-wide, as expected under the hypergeometric distribution. Dis-enrichment values are not highlighted. The phylogeny is a maximum likelihood tree based on the protein sequences of 79 genes found in single copy in most species. Key species discussed in this paper and previous literature are highlighted with blue background indicating enrichment for Mcm1 binding sites or gray indicating no enrichment. (**B**) Results of two models estimating the numbers of gains and losses of Mcm1 *cis*-regulatory sites at the RPGs. Shown is one example tree out of 10,000 sampled for each of the two models. Gains are indicated in filled circles and losses are indicated in open circles. Major nodes with a high amount of uncertainty over all of the sampled trees (0.2 < proportion with Mcm1 sites < 0.8) are shown as pie graphs with the proportion of simulations with that ancestor having Mcm1 sites shown in green. Other nodes have a high proportion (>0.8) of trees matching the example tree. (**C**) Intergenic regions upstream of two ribosomal proteins in the species *Kluyveromyces lactis* were positioned upstream of a GFP reporter. The Mcm1 *cis*-regulatory sites were scrambled and the wild-type and mutant reporters were integrated into the *Kl. lactis* genome. Cells were grown for 6 hr in rich media and expression was measured by flow cytometry. Shown is the single-cell

*Figure 1 continued on next page*

*Figure 1 continued*

fluorescence distribution for three independent genetic isolates and the median (red bar), normalized by forward-scatter values. The values were divided by the average fluorescence for a cell lacking a GFP reporter (fold above background). (D) The *RPL37* reporter strains were diluted into rich media and fluorescence and optical cell density (OD$_{600}$) were measured every 15 min in a plate reader. Shown is the change in fluorescence between consecutive time points divided by the OD$_{600}$ for eight technical replicates comprised of the three independent genetic isolates of each strain.

DOI: https://doi.org/10.7554/eLife.37563.003

The following figure supplements are available for figure 1:

**Figure supplement 1.** Evolution of RPG regulation.

DOI: https://doi.org/10.7554/eLife.37563.004

**Figure supplement 2.** Mcm1 is an activator of the RPGs in *Kl lactis*.

DOI: https://doi.org/10.7554/eLife.37563.005

the reporter (*RPS18 p*=0.017; *RPL37 p*=0.027; Welch's *t*-test) but left the cell-to-cell variability unchanged (*Figure 1C and D*; *Figure 1—figure supplement 2*). Given that there are many known regulators of ribosomal protein transcription, this demonstrates that Mcm1 plays a non-redundant role in activating these genes in *Kl. lactis* under conditions that require high expression of the translational machinery.

## Selection on RPG expression levels

Yeast ribosomes are composed of four ribosomal RNAs and 78 ribosomal proteins, assembled in equal stoichiometry (*Woolford and Baserga, 2013*). In rapidly growing cells, ribosomal protein transcripts are among the most highly expressed, and have short half-lives, leading to the estimation that approximately 50% of all RNA polymerase II initiation events occur at ribosomal proteins (*Warner, 1999*). These observations suggest that the expression of these genes is under strong selection as it plays a major role in energy expenditure in the cell.

Given that the Mcm1 *cis*-regulatory sites increase RPG transcription levels during rapid growth, there are at least two plausible hypotheses for their appearance upstream of the RPGs. (1) In those species that have acquired Mcm1 *cis*-regulatory sequences, the expression of the RPGs is higher than in species without these sequences. (2) The gain of Mcm1 *cis*-regulatory sequences compensate for other *cis*-regulatory changes that lower expression of the genes, with no net gain in expression levels. This second hypothesis is plausible, in principle, because RPGs are known to lose regulator binding sites as well as gain them (*Ihmels et al., 2005*; *Lavoie et al., 2010*; *Tanay et al., 2005*; *Tuch et al., 2008a*).

To distinguish between these hypotheses, we examined directly whether the RPGs have a higher expression level in species that have acquired Mcm1 sites compared to those that have not. (We performed these experiments under conditions in which we have shown Mcm1 sites are functional, but we do not rule out different evolutionary scenarios under other environmental conditions). To accurately measure differences in RPG expression, we mated different species pairs to form interspecies hybrids; we then measured mRNA levels by RNA-seq and assigned each sequencing read to one genome or the other (*Figure 2A*). Comparing the expression of orthologous genes in this way controls for differences in '*trans*-acting' factors like transcription regulators and therefore reflects only differences caused by *cis*-regulatory changes (*Wittkopp et al., 2004*).

For this allelic expression experiment, we constructed hybrids between *Kl. lactis* and two other species, *Kl. marxianus* and *Kl. wickerhamii*. These two additional species are relatively closely related to *Kl. lactis* (such that they can form interspecies hybrids), but have fewer Mcm1 sites at their RPGs (*Figure 2A*), suggesting that Mcm1 binding site gains are ongoing in some lineages over this timescale. mRNA reads for each gene were normalized to genomic DNA reads to control for mappability and gene length (see methods). Expression levels and differential allelic expression of genes in the two species' genomes were reproducible between replicates (*Figure 2—figure supplement 1*). At a false discovery rate of 0.05, we identified 2925 genes showing differential allelic expression out of a total of 4343 orthologs in the *Kl. lactis* × *Kl. marxianus* hybrid, and 3432 out of 4319 in the *Kl. lactis* × *Kl. wickerhamii* hybrid.

To ask broadly whether RPGs have experienced concerted *cis*-regulatory evolution, in each hybrid we asked whether the RPGs were more likely to show differential allelic expression in one direction relative to all genes in the genome (*Figure 2B*). In both hybrids, the RPGs showed evidence for *cis*-

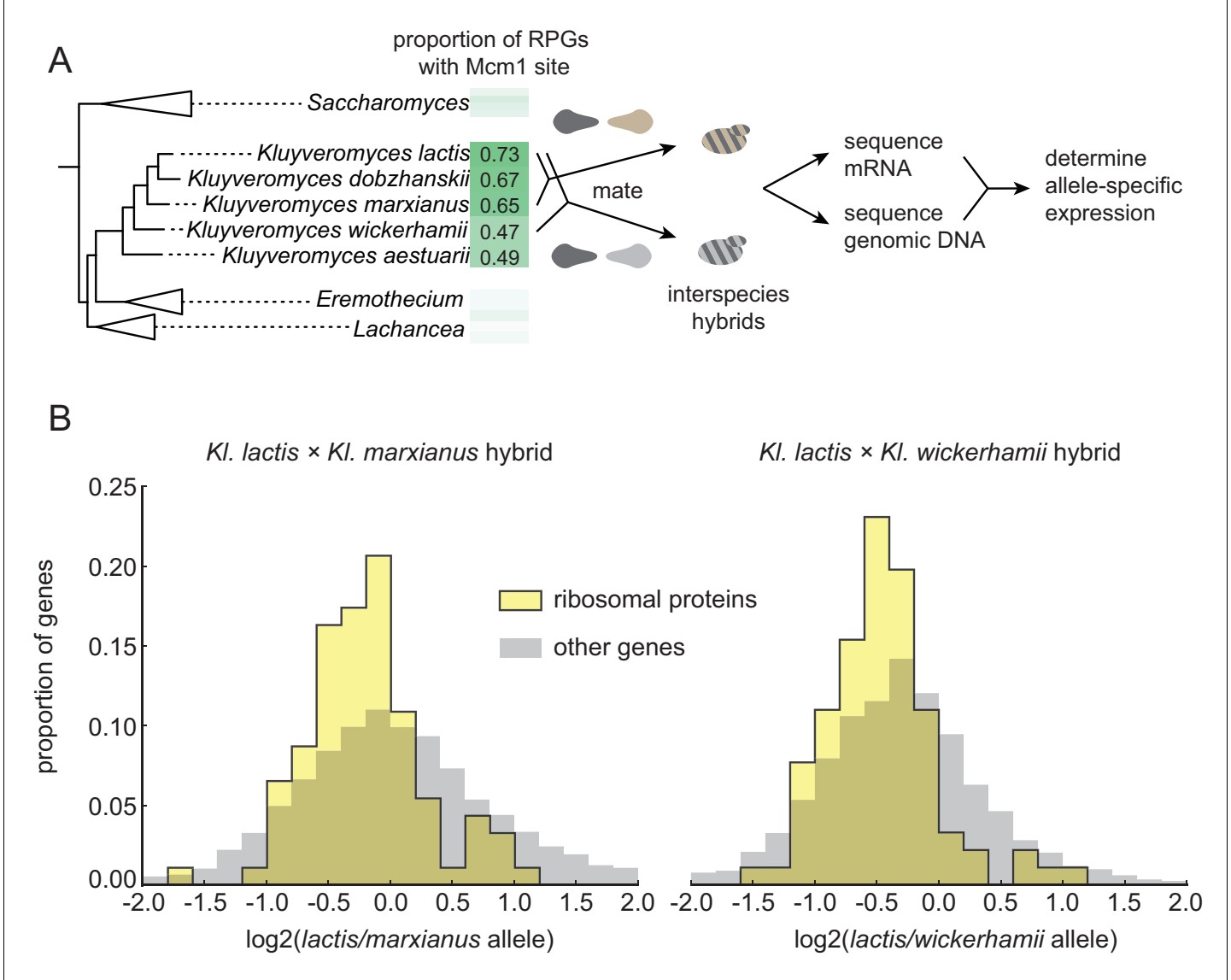

**Figure 2.** Selection on RPG expression level in *Kluyveromyces*. (**A**) Schematic of the experimental approach to measure differences in gene expression between *Kluyveromyces* yeast species. Two interspecies hybrids were constructed through mating and mRNA and genomic DNA were sequenced. The differential allelic expression is the ratio of the number of reads mapping to the coding sequence of one gene vs. its ortholog in the genome of the other species. The mRNA reads for each gene were normalized to the total reads and to the genomic DNA reads mapping to the same region to control for biases introduced in the sequencing and analysis process. (**B**) The log2-ratio of allelic expression with the *lactis* allele in the numerator is shown for (left) the *Kl. lactis* × *Kl. marxianus* hybrid (n = 3) and (right) the *Kl. lactis* × *Kl. wickerhamii* hybrid (n = 7). Shown are histograms for ribosomal protein genes and the rest of the identified orthologs in the genome.

DOI: https://doi.org/10.7554/eLife.37563.006

The following figure supplements are available for figure 2:

**Figure supplement 1.** Technical validation of allele-specific experiments.
DOI: https://doi.org/10.7554/eLife.37563.007
**Figure supplement 2.** Mcm1 site evolution and allele-specific expression.
DOI: https://doi.org/10.7554/eLife.37563.008

regulatory evolution (hypergeometric test, p=2.36e-3 for *Kl. lactis* × *Kl. marxianus* hybrid; p=1.52e-6 for *Kl. lactis* × *Kl. wickerhamii* hybrid). We conclude that in these species, the expression of the ribosomal proteins has evolved directionally as a group through *cis*-regulatory changes.

However, in both hybrids, the ribosomal protein genes were expressed, on average, lower in the *Kl. lactis* alleles than in alleles of either of the other species. This observation rules out the hypothesis that the gain of Mcm1 sites over this timescale was simply due to directional selection to increase expression of the ribosomal proteins as a whole (see also *Figure 2—figure supplement 2*). Instead, it favors the second hypothesis: ongoing evolution of Mcm1 sites in the *Kluyveromyces* clade compensates (perhaps incompletely) for other *cis*-regulatory changes that reduce RPG transcript levels. A compensatory model is consistent with the observation that the binding sites for other regulators (including Fhl1 and Rrn7) show a reduction in the *Kluyveromyces* clade (*Figure 1—figure supplement 1*).

We also revisited previously published yeast interspecies hybrid experiments performed with species from the *Saccharomyces* clade. Surprisingly, in all cases, the ribosomal proteins were among the sets of genes with reported directional *cis*-regulatory evolution (*Bullard et al., 2010*; *Clowers et al., 2015*; *Lee et al., 2013*; *Martin et al., 2012*). These observations demonstrate that RPG expression may frequently experience different evolutionary forces between closely related yeast species, and they are consistent with the high rate of gains and losses of *cis*-regulatory sequences that control RPG transcription.

## Why were Mcm1 sites gained repeatedly?

Although RPGs have experienced different selection pressures across different species, this observation does not explain why Mcm1 sites, as opposed to *cis*-regulatory sequences for many other transcription regulators, are repeatedly gained in the lineages we examined. One possibility is that Mcm1 expression changes under certain environmental conditions, conferring optimal RPG expression levels for the *Kluyveromyces* clade (as well as the other clades that gained Mcm1 sites). The fact that Mcm1 is expressed across all yeast cell types and —in combination with dedicated regulators— controls many different genes that are part of different expression programs (*Tuch et al., 2008a*) makes this possibility unlikely and we do not explicitly examine it here.

We therefore investigated whether Mcm1 has other characteristics that allow it to gain *cis*-regulatory sites especially easily at the RPGs. We previously noted that, in *Kl. lactis*, the Mcm1 sites were gained a fixed distance away from the binding site for another transcription regulator, Rap1, whose sites at the RPGs were ancestral to *Kl. lactis* and *S. cerevisiae* (*Tuch et al., 2008a*). By pooling data from additional species with Rap1-Mcm1 sites, we discovered that the spacing between Mcm1 and Rap1 sites was particularly precise with peaks at 54, 65, and 74 basepairs apart, favoring a configuration with the two proteins on the same side of the DNA helix (*Figure 3A–C*).

In order to understand whether the spacing between Mcm1 and Rap1 sites affects transcriptional activation, we moved a segment of the *Kl. lactis RPS23* upstream region that contains the Rap1 and Mcm1 sites into a heterologous reporter containing a basal promoter from *S. cerevisiae CYC1* (*Guarente and Ptashne, 1981*) (*Figure 3D*). This construct allowed us to study the Rap1 and Mcm1 *cis*-regulatory sequences independent of the additional regulators of the RPGs. We systematically varied the spacing between these two *cis*-regulatory sequences in 2 bp increments and found that the transcriptional output remained similar in most constructs (*Figure 3E*). In these experiments the Mcm1 and Rap1 sites were both close to optimal, and we hypothesized that weaker sites would be more likely to reveal a spacing preference. To test this idea, we performed a second series of experiments with the Rap1 and Mcm1 *cis*-regulatory sequences from the *Kl. lactis RPS17* gene. The binding sites from this gene are weaker matches to the Rap1 and Mcm1 position weight matrices (log odds sum of 9.34 versus 12.48 for Rap1 and 7.24 versus 14.33 for Mcm1). The transcriptional output of this second series of constructs showed sensitivity to spacing between the sites (*Figure 3E*).

We hypothesized that the observed sensitivity of transcriptional output to the strength and spacing of Rap1 and Mcm1 *cis*-regulatory sequences reflected cooperative activation of transcription. Deletion of the Mcm1 and Rap1 binding sites individually and in combination showed that the proteins *Kl*Mcm1 and *Kl*Rap1 activated transcription cooperatively in *Kl. lactis*: their combined expression was approximately four times the sum of their individual contributions (*Figure 3F*; p=5.5e-4, one sample *t*-test).

One likely explanation for these observations is the evolution of a favorable protein-protein interaction between Rap1 and Mcm1 that occurred around the time that Mcm1 sites appeared at the RPGs in the *Kluyveromyces* and *C. glabrata-Nakaseomyces* clades (*Tuch et al., 2008b*), that is a 'derived cooperativity' model (*Figure 3G*). To test this model, we placed the *Kl. lactis* Rap1-Mcm1

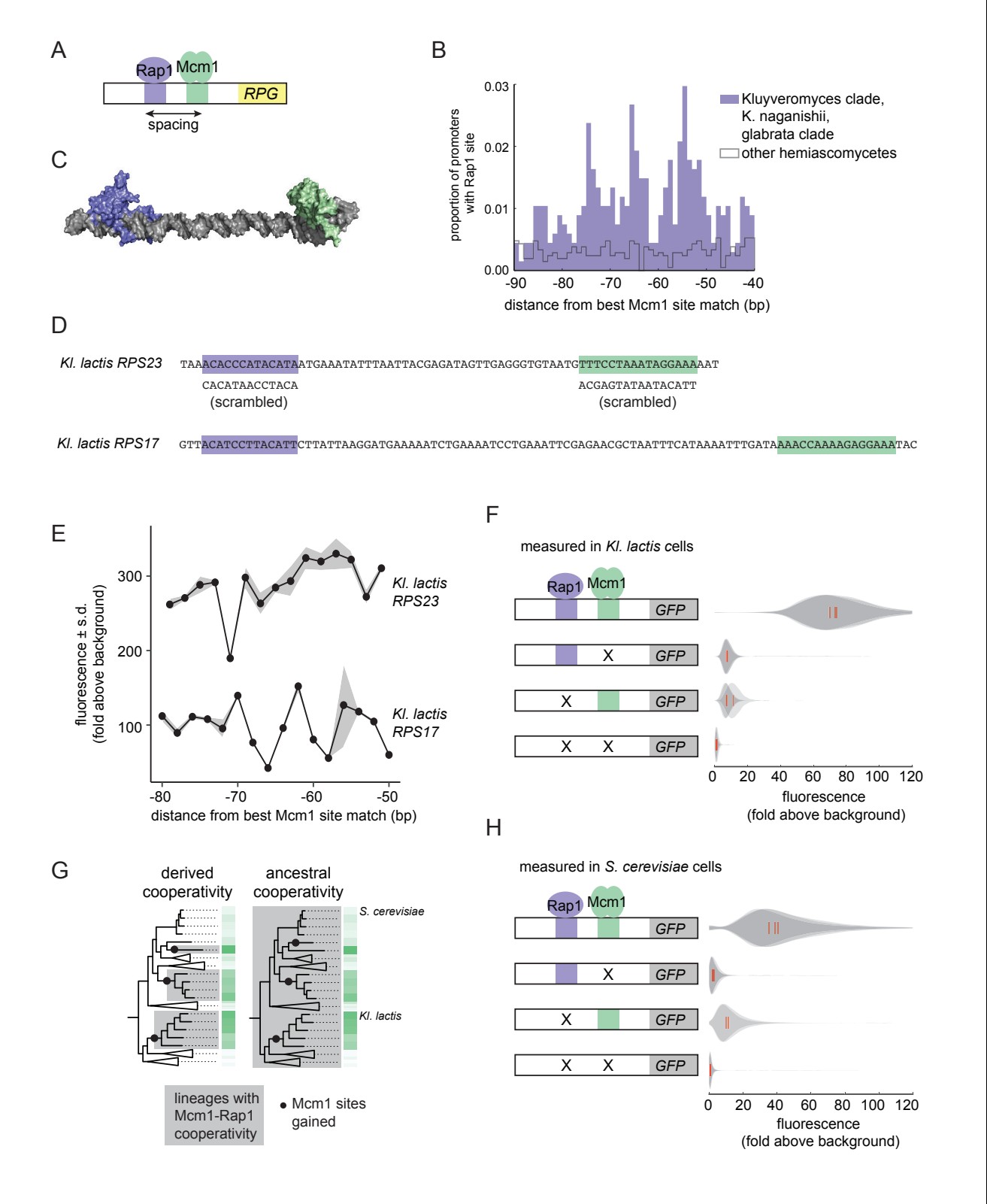

**Figure 3.** Ancestral cooperativity of Mcm1 with Rap1. (**A**) Schematic of Mcm1 and Rap1 *cis*-regulatory sites at the RPGs. (**B**) RPG promoters were aligned at the strongest hit to the Mcm1 position weight matrix and the relative location of Rap1 *cis*-regulatory sites was plotted. Sites for Rap1 with log-likelihood >6.0 are shown at 1 bp resolution. Hemiascomycete yeast (the 29 species at the top of the tree in **Figure 1A**) are divided into those with large numbers of Mcm1 sites at the ribosomal protein genes (purple shading) and those without (gray line). (**C**) Schematic using published structures of

*Figure 3 continued on next page*

*Figure 3 continued*

Mcm1 and Rap1 DNA-binding domains (PDB IDs: 1MNM and 3UKG) bound to DNA connected by a DNA linker corresponding to 55 bp spacing between their *cis*-regulatory sites. Rap1 is shown in purple and Mcm1 is shown in green. (**D–H**) The ability of Mcm1 to work with the ribosomal protein gene regulator Rap1 was tested using a GFP reporter. (**D**) The Rap1 and Mcm1 *cis*-regulatory sites from *Kl. lactis RPS23* and *RPS17* were placed in a reporter containing the *S. cerevisiae CYC1* basal promoter. Reporter variants were generated by altering the spacing between these sites and by mutating the sites individually and in combination. (**E, F**) Reporter variants were integrated into the genome of *Kl. lactis.* Cells were grown for 4 hr in rich media and expression was measured by flow cytometry. (**E**) Shown is the mean fluorescence for at least three independent genetic isolates. The values were divided by the average fluorescence for a cell lacking a GFP reporter (fold above background). The measurements for *RPS23* and *RPS17* were collected on separate days and are shown on the same axes for clarity. (**F**) Shown is the single-cell fluorescence distribution for three independent genetic isolates and the median (red bar), normalized by side-scatter values. (**G**) Diagrams showing the phylogenetic distribution of ancestral or derived cooperativity. (**H**) The *RPS23* reporter variants were integrated into the genome of *S. cerevisiae*, a species that lacks Mcm1 *cis*-regulatory sequences at the ribosomal protein genes. Cells were grown and measured as described in (**F**). (For the third construct, one isolate had multiple reporter insertions and was not included.).

DOI: https://doi.org/10.7554/eLife.37563.009

reporter into the genome of *S. cerevisiae*, a species that lacks Mcm1 sites at the RPGs. We found that *Sc*Rap1 and *Sc*Mcm1 activated expression of the reporter cooperatively (p=3.9e-3, one sample t-test), thereby demonstrating that these proteins have the capacity to work together even in a species where their binding sites did not evolve close proximity (*Figure 3H*). This result rejects the derived cooperativity model and strongly suggests that the ability of Rap1 and Mcm1 to activate transcription cooperatively was ancestral, existing even in species that did not take advantage of it in regulating the RPGs.

## Mechanism of intrinsic Mcm1-Rap1 cooperativity

Having established that cooperativity of Rap1 and Mcm1 was likely ancestral to the monophyletic clade containing *Kluyveromyces* and *Saccharomyces*, we considered three possible mechanisms that could explain ancestral cooperativity: (1) Rap1 and Mcm1 could bind DNA cooperatively through an ancient, favorable protein-protein interaction, (2) they could bind nucleosomal DNA through cooperative displacement of histones, or (3) they could both bind a third transcription regulator resulting in cooperative transcriptional activation. To test the first possibility, we used a gel-mobility shift assay with a radiolabeled DNA sequence from the *Kl. lactis RPS23* gene (*Figure 3D*). We asked whether purified full-length *S. cerevisiae* and *Kl. lactis* proteins as well as cell lysates from three additional species bound to Rap1 and Mcm1 binding sites cooperatively. In all cases, each protein bound to the RPG promoter DNA independently and did not appear to increase the other's affinity, indicating that Rap1 and Mcm1 do not, on their own, bind DNA cooperatively (*Figure 4A*, *Figure 4—figure supplement 1*).

A second mechanism explaining cooperative activation is through nucleosome displacement. Many transcription regulators have the inherent property of competing for binding to DNA with nucleosomes, with some more effective than others (*Zaret and Carroll, 2011*). However, it is unlikely that cooperative nucleosome displacement is the primary mechanism for cooperative activation by Rap1 and Mcm1 at the RPGs. Although Rap1 can indeed bind nucleosomal Rap1 binding sites (*Koerber et al., 2009*; *Rossetti et al., 2001*), and displace nucleosomes (*Kubik et al., 2015*; *Lickwar et al., 2012*; *Platt et al., 2013*), it remains bound at the RPGs even during stress conditions when the genes are repressed and show higher levels of nucleosome occupancy (*Bernstein et al., 2004*; *Lee et al., 2004*). Furthermore, a general mechanism for the cooperative assembly of all transcription regulators has little explanatory power for why *cis*-regulatory sites for Mcm1 were repeatedly gained across RPGs rather than sites for many of the ~250 other regulators coded in the yeast genome.

We next considered the third possibility, namely that the cooperative transcriptional activation we observe for Rap1 and Mcm1 occurs through the interaction of both with a third factor that catalyzes a rate-limiting step in transcription activation (*Lin et al., 1990*). In principle, a regulator could activate transcription through contacts with general transcription factors, Mediator, SAGA, various chromatin remodeling complexes, or RNA polymerase itself. To identify possible factors that might directly bind both Rap1 and Mcm1, we searched the *S. cerevisiae* BioGRID database for common interaction targets of both proteins (*Oughtred et al., 2016*). Two complexes, SWI/SNF and TFIID, fit

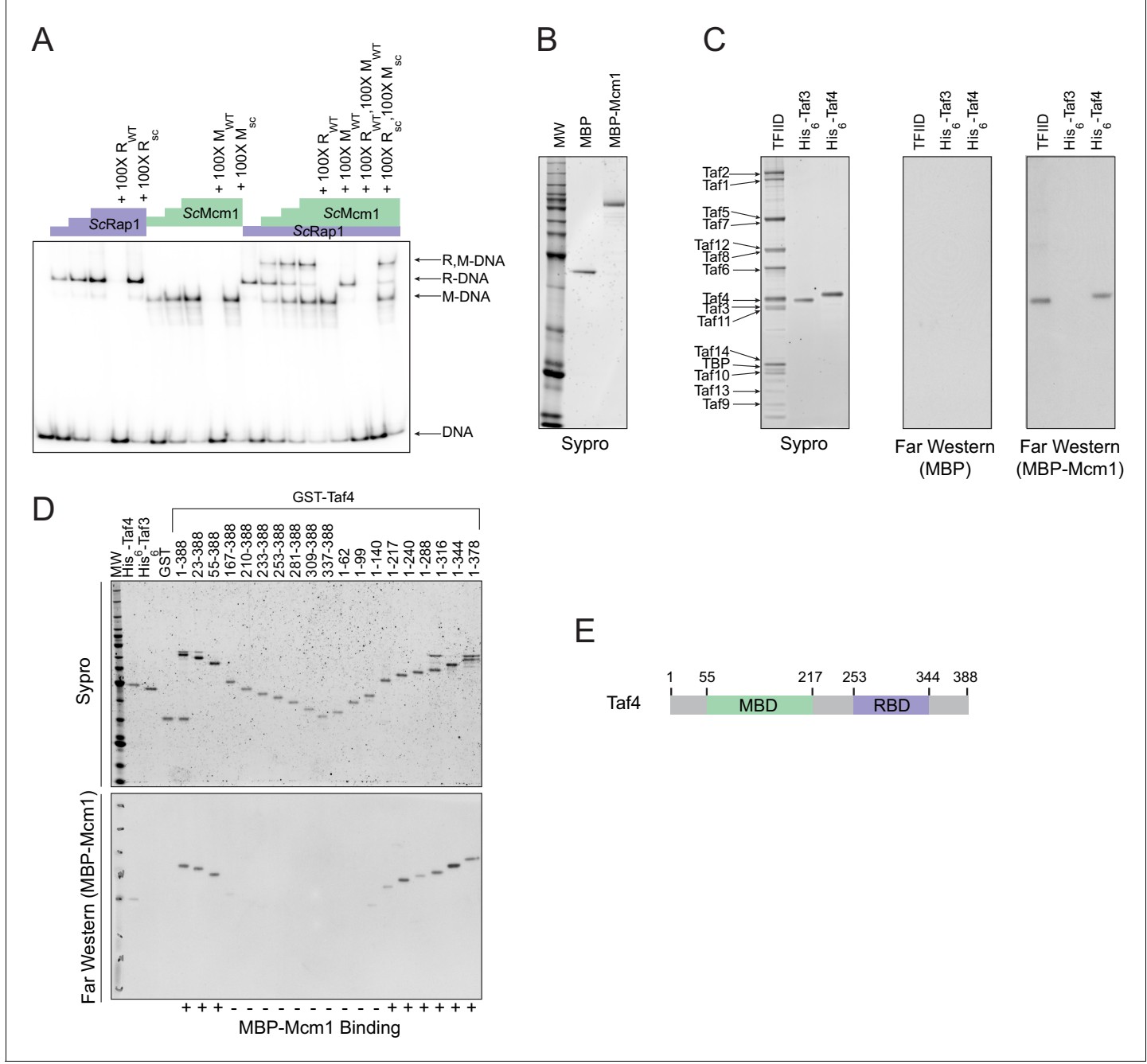

**Figure 4.** Mcm1 interacts with TFIID. Experiments were performed in *S.cerevisiae* or using *S. cerevisiae* proteins to test the mechanism of Rap1-Mcm1 cooperativity. (**A**) Gel shift DNA binding assays were performed to test possible cooperative DNA binding between purified *Sc*Mcm1 and *Sc*Rap1. Gel shift reactions were performed by incubating 10 fmol (~7000 cpm) of a 79 bp $^{32}$P-labeled fragment of the *Kl. lactis RPS23* promoter containing the Rap1 and Mcm1 binding sites (see *Figure 3D*) with either no protein, 2.5 fmol Rap1, 5 fmol Rap1, 10 fmol Rap1, 10 fmol Mcm1, 20 fmol Mcm1, 30 fmol Mcm1, or 2.5 fmol Rap1 with 10, 20, or 30 fmol Mcm1. Reactions also included either no cold competitor DNA or a 100-fold molar excess of cold ~20 bp DNA containing either the Rap1 WT ($R_{WT}$) or Rap1 scrambled ($R_{sc}$) sequences and/or the Mcm1 WT ($M_{WT}$) or Mcm1 scrambled ($M_{sc}$) sequences in a final volume of 20 µl. Reactions were fractionated on non-denaturing polyacrylamide gels, vacuum dried, and imaged using a Bio-Rad Pharos FX imager. Radiolabeled species are indicated on the left (R,M-DNA = Rap1-Mcm1-DNA, R-DNA = Rap1 DNA, M-DNA = Mcm1 DNA) (**B**) Sypro Ruby stain of SDS-PAGE fractionated MBP (2.4 pmol) and MBP-Mcm1 (1.3 pmol) probe proteins used for Far Western protein-protein binding analyses. (**C**) Far Western protein-protein binding analysis of Mcm1 binding to TFIID. Purified TFIID, His₆-Taf3, and His₆-Taf4 were separated on two SDS-PAGE gels. One gel was stained with Sypro Ruby for total protein visualization (left panel). The other was electrotransferred to a membrane for protein-protein binding analysis (middle and right panels). Membranes were probed with either control MBP (middle panel) or MBP-Mcm1 (right panel). Binding of probe proteins to Tafs was detecting using an anti-MBP antibody. (**D**) Mapping the Mcm1 Binding Domain (MBD) of Taf4. Roughly equal molar

*Figure 4 continued on next page*

Figure 4 continued

amounts of His$_6$-Taf4, His$_6$-Taf3, GST, GST-Taf4, and GST-Taf4 deletion variants were fractionated on two SDS-PAGE gels. One gel was stained with Sypro Ruby for total protein visualization. The other gel was electrotransferred to a membrane and Mcm1-Taf protein-protein binding was assayed as described in (C) using the MBP-Mcm1 as the overlay protein. (E) Taf4 protein map indicating the location of the Taf4 Mcm1 Binding Domain mapped in this study (MBD, green) as well as the Rap1 Binding Domain (RBD, purple) mapped in a previous study (*Layer et al., 2010*).
DOI: https://doi.org/10.7554/eLife.37563.010

The following figure supplement is available for figure 4:

**Figure supplement 1.** Rap1 and Mcm1 do not cooperatively bind DNA.
DOI: https://doi.org/10.7554/eLife.37563.011

this criterion. SWI/SNF is not required for ribosomal protein transcription (*Sudarsanam et al., 2000*), so it is unlikely that binding to SWI/SNF plays a role in Rap1 and Mcm1 cooperativity at the RPGs.

TFIID is a general transcription factor whose direct interaction with Rap1 is required for RPG transcription in *S. cerevisiae* (*Garbett et al., 2007*; *Layer et al., 2010*; *Mencía et al., 2002*; *Papai et al., 2010*; *Reja et al., 2015*). Furthermore, TFIID activates transcription through contacts with RNA Polymerase II, and its binding to the promoter is a rate-limiting step in the activation of TFIID-dependent genes (*Wu and Chiang, 2001*), such as the RPGs. The mechanism of how activators increase transcription rate through contacts with TFIID is not entirely understood but occurs either through a structural rearrangement of the complex or simply by an increase in its occupancy on DNA (*Coleman et al., 2017*; *Fuda et al., 2009*; *Nogales et al., 2017*; *Papai et al., 2010*; *Sauer et al., 1995a*; *Sauer et al., 1995b*).

The interaction of Rap1 with TFIID has been extensively documented (*Garbett et al., 2007*). Of particular importance is the interaction between the 'activation domain' of Rap1 and the Taf5 subunit of TFIID, as mutations that compromise this interaction strongly reduce ribosomal protein gene transcription (*Johnson and Weil, 2017*). To test directly whether Mcm1 also binds to TFIID (as suggested by mass spectrometry-based analyses of TFIID and associated proteins [*Sanders et al., 2002*]), we performed a Far-Western protein-protein binding assay using purified *S. cerevisiae* TFIID, *Sc*Mcm1, and individual TFIID subunits. We found that Mcm1 binds directly to the Taf4 subunit (*Figure 4B,C*); using deletions of Taf4, we further mapped the interaction to the N-terminal region of Taf4 (*Figure 4D,E*). Although Rap1 also binds to Taf4 (in addition to Taf5 and Taf12), its target is in the C-terminus of this subunit, distinct from the Mcm1 interaction site.

The finding that Rap1 and Mcm1 both interact with distinct domains of a common component of the transcription machinery, one whose assembly at the promoter is rate limiting, provides a simple explanation for their ability to activate transcription cooperatively. To test this idea explicitly, we performed a series of experiments in vivo. Because Rap1 is an essential gene, we took advantage of a version of Rap1 with altered DNA-binding specificity (Rap1$^{AS}$) that binds to a non-natural *cis*-regulatory sequence and confers expression of a reporter (*Johnson and Weil, 2017*). Rap1$^{AS}$ could then be manipulated without compromising the function of the endogenous Rap1. As shown in *Figure 5A–C*, Rap1$^{AS}$ shows cooperative transcriptional activation with Mcm1. When the activation domain of Rap1$^{AS}$ was mutated by introducing seven point mutations (Rap1$^{AS7Ala}$) its ability to activate transcription of a Rap1 reporter strongly decreased, but was not entirely eliminated (*Johnson and Weil, 2017*). When introduced in the presence of the Rap1-Mcm1 *HIS3* reporter, these mutations strongly reduced growth on media containing 3-aminotriazole (3-AT), a competitive inhibitor of the *HIS3* gene product (*Brennan and Struhl, 1980*), indicating that efficient expression requires an intact Rap1 activation domain (*Figure 5D–F*).

Taken together, these experiments indicate that the cooperative transcriptional activation by Rap1 and Mcm1 at the RPGs in these clades is due to both proteins interacting with TFIID, a component of the general transcriptional machinery. This idea explains how Mcm1—and not a random mixture of other regulatory proteins—came to be repeatedly gained at the ribosomal protein genes.

## Implications and predictions of the intrinsic cooperativity model

We tested several evolutionary and molecular predictions of this model. One prediction is that Rap1 and Mcm1 *cis*-regulatory sequences would occur at the prescribed distance apart but located at other genes besides the RPGs. We searched a subset of hemiascomycete yeast genomes for Rap1-Mcm1 sites with similar spacing and orientation to that observed in the RPGs (*Figure 6—figure*

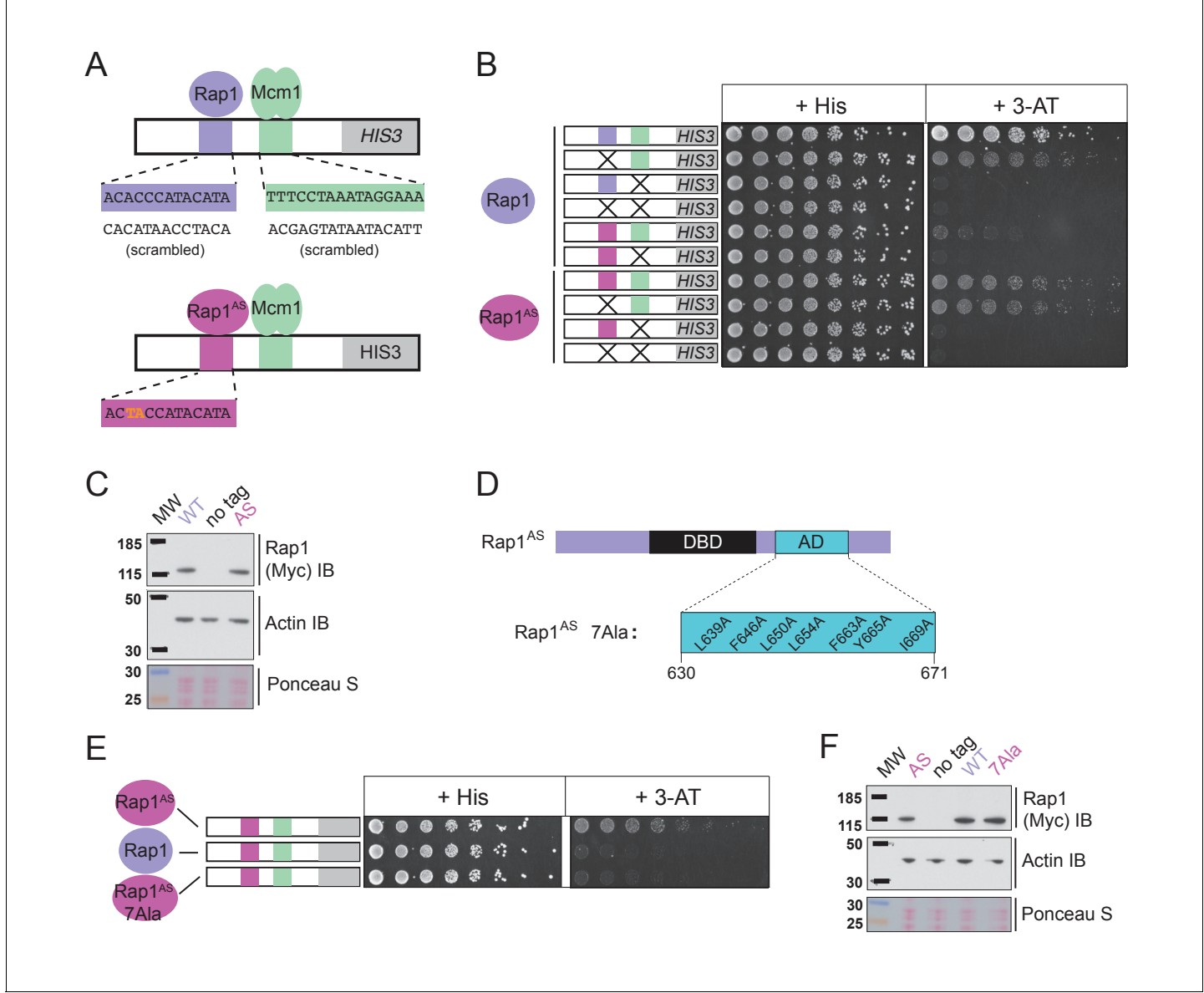

**Figure 5.** Mcm1-Rap1 cooperative activation requires Rap1-TFIID contacts. (**A**) A series of reporter constructs were designed to test the mechanism of Rap1-Mcm1 cooperative activation. Experiments were performed in *S. cerevisiae* using *S. cerevisiae* proteins. (**B**) Growth analysis of yeast strains carrying the $UAS_{Rap1\text{-}Mcm1}$ reporter (containing a fragment of the *Kl. lactis RPS23* promoter) indicated in the diagram and either an altered DNA-binding specificity Rap1 variant (Rap1$^{AS}$, magenta) or a second copy of Rap1$^{WT}$ (purple). To perform these analyses, yeast were grown overnight to saturation, serially diluted 1:4 and spotted using a pinning tool onto either non-selective media plates (+His) or plates containing 3-Aminotriazole (+3 AT), which selects for expression of the *HIS3* reporter gene. Plates were incubated for 3 days at 30° C and imaged using a Bio-Rad ChemiDoc MP imager. (**C**) Immunoblot analyses of the expression levels of Myc-tagged Rap1$^{WT}$ and Myc-tagged Rap1$^{AS}$ (Myc IB) compared to actin (Actin IB) and total protein (Ponceau S) loading controls. (**D**) Rap1 protein map indicating the *Sc*Rap1 AD mapped to a location C-terminal of the Rap1 DBD and the seven key AD amino acids. These amino acids were mutated to alanine to inactivate the *Sc*Rap1 AD and create the Rap1$^{AS}$ 7Ala mutant variant. (**E**) Growth analyses performed using yeast carrying the $UAS_{Rap1AS\text{-}Mcm1}$-*HIS3* reporter and either Rap1$^{AS}$, a second copy of Rap1$^{WT}$, or Rap1$^{AS7Ala}$ performed as described in 'B.' (**F**) Immunoblot analysis of the Rap1 forms tested in (**E**) performed as described in (**C**).

DOI: https://doi.org/10.7554/eLife.37563.012

supplement 1). Most species analyzed had only a few such genes of questionable significance, but *Ka. naganishii* had 33, showing that Rap1 and Mcm1 sites can evolve at genes other than the RPGs.

A second prediction of our model is that even a sub-optimal Mcm1 site would activate transcription of a regulatory region with an ancestral, strong Rap1 site. Because sub-optimal sites can evolve

de novo with much higher probability than optimal sites (*Dermitzakis and Clark, 2002*; *Stone and Wray, 2001*), this property would increase the ease by which a large number of functional Mcm1 sites could arise during evolution. To test this prediction, we created a series of GFP reporter constructs with Mcm1 binding sites that drive different levels of expression (*Acton et al., 1997*). We tested the expression level of each of these Mcm1 sites in the presence and absence of a neighboring Rap1 site in *Kl. lactis* (*Figure 6A*). Consistent with the prediction, most of the sub-optimal Mcm1 sites displayed near wild-type levels of expression in the presence—but not in the absence—of neighboring Rap1 sites.

Finally, we asked whether our conclusions are generalizable to other pairs of transcription factors besides Mcm1 and Rap1. In particular, some of the clades identified in *Figure 1A* gained Mcm1 sites at RPGs that lack Rap1 sites. To address this question, we centered each RPG intergenic region on the best Mcm1 site and plotted the location of other RPG regulators relative to Mcm1 (*Figure 6B*). This analysis revealed that, in some of these clades, Mcm1 sites were gained varying distances away from the pre-existing sites of other regulators (Tbf1, Rrn7, and Fhl1), suggesting that the evolutionary mechanism we identified with Rap1 and Mcm1 might be a generalizable to other instances of *cis*-regulatory evolution (*Figure 6C*). Consistent with this idea, all three of these regulators are reported to interact (directly or indirectly) with TFIID (*Gavin et al., 2002*; *Knutson et al., 2014*; *Mallick and Whiteway, 2013*; *Zhong and Melcher, 2010*).

In summary, the experiments we have presented describe a special relationship between Rap1 and Mcm1 by virtue of their interaction with different surfaces of the same rate-limiting component of transcription, TFIID. This relationship between Rap1 and Mcm1 is ancient, and in the next section, we discuss how this property can predispose transcription networks to evolve repeatedly along the same trajectory.

## Discussion

Here we have investigated an example of parallel evolution where binding sites for a particular transcription regulator (Mcm1) were gained in a large group of genes (the ribosomal protein genes, RPGs). These gains occurred repeatedly in several independent fungal lineages. In three of these lineages, Mcm1 binding sites were gained a fixed distance from the sites for another transcription regulator, Rap1, and we show that these newly acquired Mcm1 sites are required for full activation of the RPGs. We also show that Mcm1 and Rap1 cooperatively activate these genes. The direct interaction of both of these regulators with a common target, the general transcription factor TFIID, provides a plausible mechanism for this cooperative transcription activation. It also explains why the ability of Rap1 and Mcm1 to work together was ancestral to the more recent gains of Mcm1 sites adjacent to Rap1 sites at the RPGs.

How do these observations account for the fact that Mcm1 sites (as opposed to sites for other regulators) were repeatedly gained in parallel next to the Rap1 site at the RPGs? And how do they account for the distance constraints? One common explanation for parallelism is a specific environmental adaptation that occurs through a similar molecular mechanism. However, the yeast species with Mcm1 sites at the RPGs are from diverse ecological niches (e.g. plant leaves, mangrove sediment, the human body, soil) and utilize different nutrient sources (e.g. lactate, xylose, feline skin, nematode predation), defying a specific environmental adaptation explanation. Consistent with this view is our observation, based on analyzing interspecies hybrids, that a species in which the Mcm1 sites were gained at the RPGs does not express these genes at higher levels than related species that evolved fewer Mcm1 sites.

The model that best fits all of our data holds that the parallel gains arose from the ease with which the functional Mcm1 sites (and not the sites of other regulators) appeared in evolution, rather than selective pressure for particular adaptation. Specifically, we propose that, in the *Kluyveromyces-Saccharomyces* ancestor (before the parallel gains of Mcm1 sites) Rap1 bound to the ribosomal protein genes and activated transcription through interactions with TFIID, as it does in extant species. We further propose that, in the ancestor, Mcm1 activated non-ribosomal genes by interaction with a second site on TFIID, as it does in the extant species *S. cerevisiae*. Any suboptimal Mcm1 site that arose by chance point mutation at a specified distance from a Rap1 site would immediately be functional (*Figure 7*) because, as we show, even a weak Mcm1 DNA interaction would be stabilized by Mcm1's intrinsic ability to directly bind TFIID (see *Figure 4C and 6A*). In this way, even

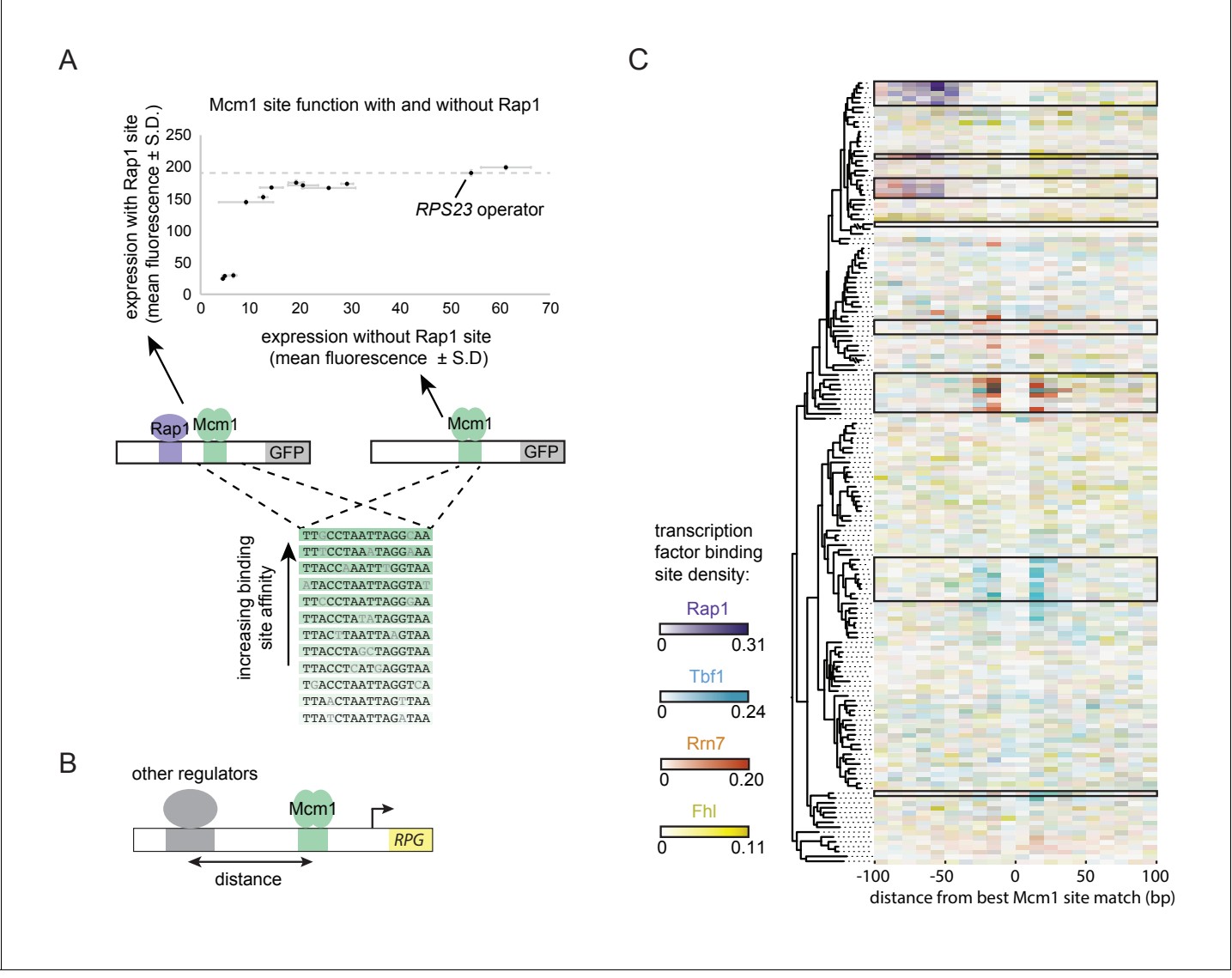

**Figure 6.** Evolutionary implications of intrinsic cooperative activation. (A) A series of reporters were designed to test the transcriptional activation of a weak Mcm1 site in the presence and absence of a Rap1 site. A series of Mcm1 *cis*-regulatory sites were chosen with a range of affinities that correlate with transcription rate (***Acton et al., 1997***). The order of the sequences shown corresponds to their expression level on the x-axis. These sites were introduced to the *S. cerevisiae CYC1* reporter and tested with (y-axis) and without (x-axis) an upstream Rap1 binding site. Cells were grown and measured as described in *Figure 3E*. The expression level of the WT *RPS23* operator is shown as a dotted line. (B) A computational analysis was designed to detect evolution of Mcm1 sites at fixed distances from other ribosomal protein regulators. Ribosomal protein gene promoters were aligned at the strongest hit to the Mcm1 position weight matrix and the relative location of *cis*-regulatory sites for other transcription regulators was plotted. (C) The shading in each rectangle represents the proportion of ribosomal protein gene promoters in that species that have the given *cis*-regulatory site in that 10 bp interval. The clades with a large number of Mcm1 *cis*-regulatory sequences are shown with black boxes.

DOI: https://doi.org/10.7554/eLife.37563.013

The following figure supplements are available for figure 6:

**Figure supplement 1.** Evolution of Rap1-Mcm1 sites at additional genes.
DOI: https://doi.org/10.7554/eLife.37563.014

**Figure supplement 2.** Evolution of Mcm1 *cis*-regulatory sites near sites for other regulators.
DOI: https://doi.org/10.7554/eLife.37563.015

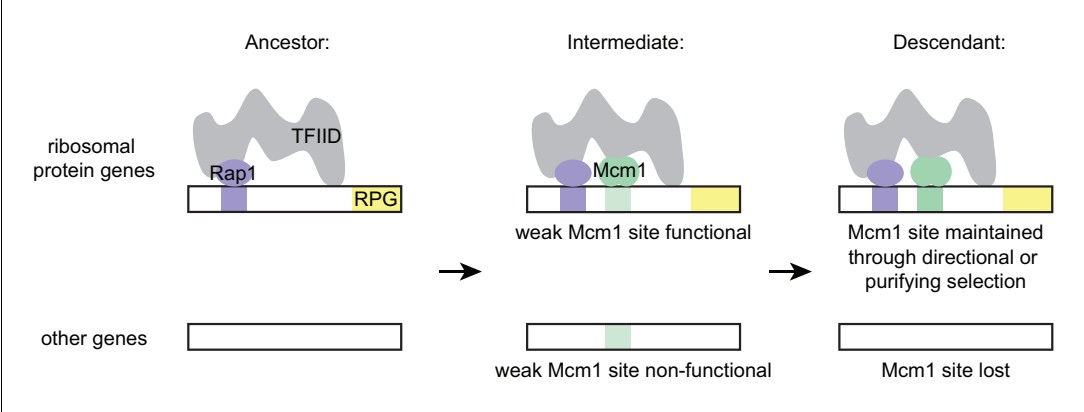

**Figure 7.** Model for evolution of *cis*-regulatory sites through intrinsic cooperativity. Multiple gains of new Mcm1 sites occur in the ribosomal protein genes because Rap1 and Mcm1 both bind to TFIID, a general transcription factor. Due to the intrinsic cooperativity of Rap1 and Mcm1 (which is ancestral to the gains of Mcm1 *cis*-regulatory sequences) the evolution of even a weak Mcm1 site near an existing Rap1 site would produce an effect on transcription. Because they are more likely to be functional, weak Mcm1 *cis*-regulatory sequences are preferentially retained in the population if they arise at a specified distance (as determined by the shape of TFIID) from Rap1 *cis*-regulatory sequences. These sites would be preserved if there is direct selection to increase RPG expression, or if they are combined over time with the mutational degradation of other regulatory elements that bring the Mcm1 site under purifying selection.
DOI: https://doi.org/10.7554/eLife.37563.016

suboptimal *cis*-regulatory sequences (which are much more likely to appear by chance than optimal sites [***Dermitzakis and Clark, 2002***; ***Stone and Wray, 2001***]) could form under selection. The appearance of Mcm1 sites likely occurred concomitantly with the gradual losses of other *cis*-regulatory sites in the RPGs; in other words, the Mcm1 *cis*-regulatory sites would fall under selection as other *cis*-regulatory sites deteriorated by mutation. In essence, we propose that the free energy gained from the intrinsic interaction between Mcm1 and TFIID would favor formation of new Mcm1 sites at the expense of pre-existing *cis*-regulatory sequences, particularly since the latter provide a larger target for inactivating mutations. This model accounts for why Mcm1 sites (and not those of other transcription regulators) were repeatedly gained at the RPGs and why the distance between Rap1 and Mcm1 sites is constrained in those species in which the gains occurred.

Numerous experimental observations support this model and rule out alternative explanations: (1) in extant species, Rap1 and Mcm1 both interact with TFIID; (2) They interact with different parts of TFIID; (3) cooperative transcriptional activation by Mcm1 and Rap1 requires the activation domain of Rap1, which is known to interact with TFIID; (4) the spacing between Rap1 and Mcm1 sites in the ribosomal protein genes places the proteins on the same side of the helix but at least 50 bp apart, consistent with a physical interaction with a large complex; (5) engineered suboptimal Mcm1 sites are functional as long as they are adjacent to Rap1 sites; (6) Mcm1 and Rap1 have the intrinsic ability to cooperate (through interactions with TFIID) even in a species where Mcm1 sites were not gained at the RPGs.

We note that this model does not require any change in Rap1 or Mcm1 during the gains of Mcm1 sites at the RPGs. Presumably, Rap1 and Mcm1 activated many genes separately in the ancestor, thus preserving by stabilizing selection their ability to interact with TFIID. This idea is consistent with the observation that the two proteins were able to cooperate on artificial constructs introduced into *S. cerevisiae* even though their binding sites are not found together at the RPGs. Mutations that alter the function of transcription regulators (for example, creating a new protein-protein interaction) can be pleiotropic, decreasing the likelihood that they can arise without disrupting the proteins' ancestral functions (***Carroll, 2005***; ***Stern and Orgogozo, 2008***). (We note that the transcriptional output was slightly more cooperative in *Kl. lactis* than in *S. cerevisiae*, leaving open the possibility that additional fine-scale evolutionary changes may have occurred in how these proteins interact.) According to our model, the ability of the two regulators to work together was ancestral, part of each protein's intrinsic mechanism of transcriptional activation; therefore, their coupling at the RPGs would avoid such pleiotropic changes.

How does this model account for the gains of Mcm1 sites observed in clades where Rap1 does not regulate the RPGs? In these cases, Mcm1 *cis*-regulatory sequences also show preferred spacing relative to known regulators of the RPGs, specifically Tbf1 and Rrn7, and we propose that the same type of cooperativity with TFIID can also account for these cases. These two regulators are reported to interact with TFIID (*Knutson et al., 2014*; *Mallick and Whiteway, 2013*). Indeed, TFIID occupies the promoters of RPGs in human cell lines as well (*ENCODE Project Consortium et al., 2012*), raising the possibility that TFIID is a conserved general activator of the RPGs across fungi and animals, while the specific transcription regulators that interact with TFIID simply interchange over this timescale.

While our cooperative activation model provides an explanation for the parallel acquisition of Mcm1 *cis*-regulatory site evolution, selection must have operated to preserve the Mcm1 sites as they arose in the population. As described earlier, we favor a model where the gains of Mcm1 sites compensated for the degradation of other *cis*-regulatory sequences and thereby fell under selection. The allelic expression data (*Figure 2*) strongly supports this model for one clade, represented by *Kluyveromyces* species. However, it is also possible that, in other clades or over shorter timescales, the Mcm1 sites could have been gained due to selection for higher levels of RPG expression. The widespread differences in RPG expression revealed by the published interspecies hybrid experiments suggest that RPG expression may experience strong and shifting selection, helping to account for the surprising observation that transcriptional regulators that control the RPGs vary substantially across species (*Gasch et al., 2004*; *Lavoie et al., 2010*; *Mallick and Whiteway, 2013*; *Tanay et al., 2005*; *Tuch et al., 2008a*; *Zeevi et al., 2014*; *Zeevi et al., 2011*). We note that the intrinsic cooperativity model is sufficient to explain the gain of Mcm1 sites whether or not any change in selection occurs: if functional Mcm1 sites are relatively easy to form (because of intrinsic cooperativity) they will be favored over gains of other *cis*-regulatory sequences by mutational processes alone. Mutations creating Mcm1 sites and weakening other sites could occur sequentially in either order or simultaneously (on the same haplotype), depending on the expression requirements of the RPG at that point in its evolutionary history. We feel that it is likely that most or all of the selection scenarios outlined above have occurred in at least one RPG at some point during the multiple and ongoing gains of Mcm1 sites over millions of years of fungal evolution. While the precise circumstances of each Mcm1 site gain are unknown, intrinsic cooperativity biases the RPGs as a whole toward gaining these sites.

In conclusion, we have proposed a mechanism, supported by multiple lines of experimental evidence, to account for convergent regulatory evolution of a large set of genes. Although, on the surface, the parallel gains of Mcm1 sites at the ribosomal genes would seem to require a special evolutionary explanation, our model does not require an extraordinary mechanism beyond individual point mutations in the *cis*-regulatory region of each gene. However, the ancestral ability of the two key regulators to activate transcription simplifies the path to gaining these sites by producing a phenotypic output from even non-optimal sites. Mcm1, because of its intrinsic ability to cooperate with Rap1, can significantly activate transcription at the ribosomal proteins more easily than it would elsewhere in the genome; likewise, Mcm1 (or another regulator that interacted with TFIID) would be preferred at the RPGs over regulators that did not share this common direct protein interaction. Thus, the intrinsic cooperativity of Rap1 and Mcm1 'channels' random mutations into functional Mcm1 *cis*-regulatory sequences, accounting for the observed parallel evolution. We speculate that gene activation through intrinsic cooperativity may be a general mechanism to explain the rapid and ubiquitous rewiring of transcription networks.

## Materials and methods

### Computational genomics of *cis*-regulatory sequences

Genomes were compiled from the Yeast Gene Order Browser (YGOB) (*Gordon et al., 2011*), the Department of Energy Joint Genome Institute MycoCosm portal (*Grigoriev et al., 2014*), and individual genome releases (*Supplementary file 3*). The *Kl. wickerhamii*, *Kl. marxianus*, and *H. vinae* genomes were annotated using the Yeast Gene Annotation Pipeline associated with YGOB. The annotations of genes and proteins in other genomes were obtained from the source of the genome sequence. RPGs were defined as any gene starting with 'Rps' or 'Rpl' in *S. cerevisiae* and were

identified using the ortholog annotation in YGOB (for species contained in this repository). These genes were identified in genomes from other sources through psi-blast in BLAST+ (*Camacho et al., 2009*), based on their high conservation.

The species tree was created as described (*Lohse et al., 2010*). In total, 79 single-copy orthologs were chosen to create the species tree based on the ortholog mapping repositories YGOB and Fungal Orthogroups (*Byrne and Wolfe, 2005*; *Wapinski et al., 2007*). They were aligned individually using MUSCLE with default parameters (*Edgar, 2004*). For species that were not included in these repositories, the corresponding ortholog was identified using psi-BLAST (*Altschul et al., 1997*). Using the orthologs from every species, the genes were then re-aligned using MUSCLE with default parameters and concatenated into a single alignment. To calculate the tree topology, FastTree 2.1.8 was used with Blosum45 matrix, the JTT model with 20 rate categories, and two rounds of +NNI +SPR (default parameters).

Intergenic regions were extracted upstream of each gene using the python script intergenic.py. For scoring of potential transcription factor binding sites, motifs were obtained from the ScerTF database (*Spivak and Stormo, 2012*) as position weight matrices. Each intergenic region (in the ribosomal proteins or genome-wide, depending on the question) was scored using the script TFBS_score.py by adding up the log likelihood values of each base at each position in the motif, forward and backward, and then repeating for each position in the intergenic region. It is important to note that the motifs from ScerTF are corrected for the GC-content of the *S. cerevisiae* genome (or the genome(s) from which the matrix is derived), but not individually for the genomes of each species in other parts of the tree. The motif was left as-is instead of correcting for the GC-content in each genome, because the purpose of this scoring was to identify DNA sequences that are most similar to the Mcm1 binding site, not those that are most statistically enriched given the GC-content. Calculating enrichment of the binding site in ribosomal protein genes relative to the rest of the genome was done to take into account forces (including, but not limited to GC-content) that affect the prevalence of the motif genome-wide. After determining the conclusions were unchanged when using 500, 1000, and 1500 bp for the maximum length of an intergenic region, the length of 1000 bp was chosen. A cutoff was chosen for presence of the *cis*-regulatory site (*Figure 1—figure supplement 1*).

Ribosomal protein gene intergenic regions were also screened for motifs that were not present or were more information-rich than the corresponding motif in the ScerTF database. This was done by querying the intergeneic regions in each species using MEME with the zero-or-one occurrence per sequence option. From these results the following motifs were chosen: Cbf1 from *Spathaspora passalidarum*, Rrn7 from *Arxula adeninivorans*, Tbf1 from *Ascoidea rubescens* and *Schizosaccharomyces japonicus*, Hmo1 from *Lachancea thermotolerans* and a widespread but previously unidentified motif from *Botrytis cinerea*. The Dot6, Fhl1, Rap1, Mcm1, Rim101, Sfp1, and Stb3 motifs were obtained from ScerTF.

To estimate the number of gains and losses, species were categorized based on whether their RPGs contained Mcm1 sites (-log10($P$) > 3) or not. We used phytools to simulate 10,000 stochastic character maps under the equal rates model (ER) and a different-rates model (ARD) assuming Mcm1 sites were a discrete character (*Bollback, 2006*; *Revell, 2012*). These trees were used to determine the number of gains and losses of Mcm1 sites under each model and find which ancestral nodes were likely to have Mcm1 sites at the RPGs and which were uncertain.

Previous studies have indicated that many features of the RPGs are defined by their relative location to the Rap1 binding site (*Reja et al., 2015*). To identify the relative locations, the best hit for the Rap1 site in each species was identified, then a cutoff was set (approximately half of the maximum potential score for a given position weight matrix) for the motif of the second transcription factor. The location of a binding site was defined as the midpoint of the motif. This analysis was carried out using the scripts TFBS_score.py and rel_locs_RPs.py.

Genome-wide scoring of relative motif locations using rel_loc_RPs.py was used to identify additional genes beyond the RPGs that show a similar pattern of Rap1 and Mcm1 sites near to each other. Rap1 sites that faced toward the gene and had a score greater than 6.0 were identified, as were Mcm1 sites with a score above 6.0. Then, genes that had both a forward-facing Rap1 site and an Mcm1 site between 52 and 78 bp downstream (the spacing of most sites at the RPGs) were identified. The order of Rap1 and Mcm1 *cis*-regulatory site appearance in evolution was inferred by the distribution of the sites in closely related species with available genome sequences.

## Strain and reporter construction

The GFP and *HIS3* reporters used in *K. lactis* and *S. cerevisiae* have been previously described (*Garbett et al., 2007*; *Mencía et al., 2002*; *Sorrells et al., 2015*). These reporters allow full intergenic regions to be cloned upstream of GFP or *HIS3*, with the break between the original gene sequence and the reporter gene occurring at the start codon. A second version of the GFP reporter uses the *CYC1* promoter from *S. cerevisiae* (*Guarente and Ptashne, 1981*) with its upstream activation sequence replaced by two restriction enzyme sites. Different versions of these vectors were made to integrate into the *K. lactis* genome and into the *S. cerevisiae* genome. Plasmids and strains are reported in *Supplementary files 1* and *2*.

To make the full-length RPG GFP reporters, the wild-type intergenic regions were obtained by PCR with ExTaq (Takara) from genomic DNA, flanked by directional restriction enzyme sites for SacI and AgeI. These were cloned using a 2:1 ratio into pTS16 digested with the same restriction enzyme sites, and ligated using Fastlink ligase (Epicentre) to make pTS170 and pTS174.

To scramble Mcm1 and Rap1 binding sites, these sites were put into a text scrambler, then the resulting sequences were queried in ScerTF (*Spivak and Stormo, 2012*) to see if they contained matches to any other known transcription factor binding site motifs. If not, they were used for further experiments.

To make pTS171, pTS175, and pTS243-246 DNA sequences were synthesized containing scrambled Mcm1, Rap1 sites, or both. The insert for pTS171 and pTS243-244 were cloned into pTS170 using the restriction enzymes SacI and Bsu36i. The insert for pTS175 and pTS245-246 was cloned into pTS174 using SacI and EcoRV.

The vectors pTS176-179 contain the *K. lactis RPS23* Rap1-Mcm1 operator upstream of the *CYC1* promoter. These vectors were made by annealing oligos and ligating them in a 50:1 ratio into pTS26 digested with NotI and XhoI. The equivalent vectors for *S. cerevisiae* are pTS181-184 and were made by cloning into pTS180 with NotI and XhoI. The constructs testing the spacing between Rap1 and Mcm1 sites (pTS189-203 and pTS209-224) were cloned using the same approach. For pTS189-203 the intervening sequence between the sites was partially duplicated for some constructs to increase the spacing. For pTS209-224 the endogenous spacing was 80 bp so the entire series was made with deletions starting immediately downstream from the Rap1 site.

The reporters testing how weak Mcm1 sites cooperate with Rap1 were cloned by first adding a BamHI site along with a palindromic Mcm1 site (*Acton et al., 1997*) into the reporter containing the *K. lactis RPS23* Rap1-Mcm1 operator to make pTS247 and pTS248 (with a scrambled Rap1 site). Then variants of the palindromic site containing point mutations were cloned into these two vectors using BamHI and XhoI. For each variant, two point mutations were made to preserve the palindromic nature of the Mcm1 binding site (*Acton et al., 1997*).

These reporters were digested with KasI and HindIII and integrated into the *K. lactis* genome by transformation as previously described (*Kooistra et al., 2004*) and into the *S. cerevisiae* genome using a standard lithium-acetate transformation. Yeast were grown on non-selective media for 24 hr then replica plated onto plates containing at least 100 µg/mL Hygromycin B. For the spacing and weak Mcm1 reporter series, the reporters are enumerated in the plasmid list but not the strain list. This is because they were transformed into *K. lactis*, tested, then discarded due to their large numbers. For each reporter, four independent isolates were measured, but isolates where the full reporter had not integrated were discarded, resulting in 3 or four replicates per construct.

The *HIS3* reporters were generated by performing PCR on the equivalent GFP reporters to generate wild-type, Rap1[AS], and scrambled versions of the *RPS23* fragment containing Rap1 and Mcm1 binding sites with NcoI and SacII restriction sites on the ends. These fragments were cloned in a 3:1 ratio into similarly digested $UAS_{Rap1WT}$-*HIS3* reporter plasmid (*Garbett et al., 2007*; *Mencía et al., 2002*). These reporters were digested with SpeI and SalI and integrated into the *S. cerevisiae* genome via lithium-acetate transformation. Transformants were selected on media lacking *TRP1* and integration at the correct locus was confirmed via PCR.

## GFP reporter assays

*K. lactis* and *S. cerevisiae* reporters were grown overnight in 1 mL cultures in 96 well plates in synthetic complete media. The next day, cells were diluted into synthetic complete media to $OD_{600}$ = 0.025 – 0.05 and grown for 3 hr. Cells were measured by flow cytometry on a BD LSR II

between 3 hr and 4 hr after dilution. A total of 10,000 cells per strain were recorded. Cells were gated to exclude debris, and the mean fluorescence for each strain was used for comparing among different strains. For each reporter, three to four independent isolates were checked, as we were interested in large differences and standard deviations within samples were small. In the case that one of the isolates anomalously showed expression equivalent to background, while the other isolates showed similar but detectable fluorescence, the anomalous isolate was excluded from analysis. Experiments were performed a minimum of two times on different days. Strains were not blinded for data collection or analysis.

### HIS3 reporter assays

HIS3 reporter expression in *S. cerevisiae* was scored by growth assays performed on three independent biological replicates. In these assays, *S. cerevisiae* were grown overnight to saturation and serially diluted 1:4 in sterile water in 96-well plates. These dilution series were spotted using a pinning tool (Sigma) onto Synthetic Complete (SC) media (0.67% (w/v) yeast nitrogen base without amino acids, 2% (w/v) dextrose, 0.2% (w/v) amino acid dropout mixture) either with His (+His, non-selective media) or without His and with 3-aminotriazole (-His + 3 AT, selective media). Plate images were acquired using the ChemiDoc MP Imager (Bio-Rad) and processed using ImageLab software (Bio-Rad) after 2 days of growth at 30°C.

### Interspecies hybrids

To construct the interspecies hybrids for allelic expression measurements, multiple isolates of different species were mated together. Strains with complementary markers were grown on YEPD plates for 2 days, then mixed together in patches on 5% malt extract plates for 2 – 4 days. These patches were then observed under the microscope to check for zygotes and streaked out onto plates that select for mating products. Hybrids were tested by PCR for products that were species, and mating-type specific. *Kl. lactis* × *Kl. dobzhanskii* matings were attempted, but were unsuccessful, perhaps because the *Kl. dobzhanskii* isolate used was an **a**/α strain. *Kl. lactis* × *L. kluyveri* zygotes were observed when the two species were mixed with *Kl. lactis* alpha-factor, but no mating products were obtained. One cross of *Kl. lactis* and *Kl. aestuarii* produced zygotes and mating products, but the *Kl. aestuarii* isolate turned out to be an isolate of *Kl. wickerhamii* instead (discovered upon genome-sequencing). In all, three *Kl. lactis* × *Kl. wickerhamii* matings, and one *Kl. lactis* × *Kl. marxianus* mating—each with three isolates—were obtained and carried forward for analysis. *Kl. lactis* × *Kl. wickerhamii* hybrids are yTS347 and yTS349 (two matings between yLB13a and yLB122), and yTS353 (a mating between yLB72 and yLB66c). The *Kl. lactis* × *Kl. marxianus* mating was yTS352 (a mating between yLB72 and CB63). Sample sizes were determined by the number of samples in our sequencing kit and the number of isolates we recovered from matings.

Both mRNA and genomic DNA were sequenced from the hybrids. Genomic DNA of one isolate of each of the *Kl. lactis* × *Kl. wickerhamii* hybrids (yTS347, yTS349 and yTS353) was sequenced, along with all three isolates of the *Kl. lactis* × *Kl. marxianus* hybrid, and one isolate of each of the parental strains. Cells were grown in 5 mL cultures overnight in YEPD. Genomic DNA was prepared using a standard 'smash and grab' protocol, where cells are lysed with phenol/choloroform/isoamyl alcohol and glass beads. DNA was precipitated twice and treated with RNase A, then sheared on a Diagenode Bioruptor for 2 × 10 min (30 s on 1 min off) on medium intensity. Genomic DNA was prepared for sequencing using the NEBNext Ultra DNA Prep Kit for Illumina E7370 (New England BioLabs).

All three isolates of each of the four hybrids were prepared for mRNA sequencing. Cells were grown overnight, then diluted back into YEPD to $OD_{600}$ = 0.2 and grown for 4 – 8 hr until they reached $OD_{600}$ = 0.7 – 1.0. The growth rate of the *Kl. lactis* × *Kl. wickerhamii* hybrids was slower and more variable between isolates. At this point, cells were pelleted and frozen in liquid nitrogen. mRNA was extracted using the RiboPure kit AM1926 (Applied Biosystems). Polyadenylated RNAs were selected using two rounds of the Oligotex mRNA Kit 70022 (Qiagen). The samples were then concentrated using the RNA Clean and Concentrator-5 (Zymo Research).

Libraries were prepared using the NEBNext Ultra Directional RNA Library Prep Kit for Illumina E7420 (New England BioLabs). mRNA and library quality were assessed using a Bioanalyzer 2100

(Agilent). Sequencing was performed at the University of California, San Francisco Center for Advanced Technology on an Illumina HiSeq 4000.

## Allelic expression analysis

Raw sequencing data was checked for quality control using FastQC (http://www.bioinformatics.bab-raham.ac.uk/projects/fastqc/). Next, each of the genomic DNA isolates was aligned separately to each of four genomes: *Kl. lactis*, *Kl. marxianus*, *Kl. wickerhamii*, and *Kl. aestuarii*. In each case, reads uniquely mapped to the expected genome(s) and few reads mapped to other genomes. The strain yTS349 was previously thought to be a *Kl. lactis* ×*Kl. aestuarii* hybrid, but sequencing revealed it was in fact a *Kl. lactis* ×*Kl. wickerhamii* hybrid, and was treated as such for the analyses.

The genomes for *Kl. lactis*, *Kl. marxianus*, and *Kl. wickerhamii* were annotated using the yeast genome annotation pipeline from YGOB to standardize the gene annotation and ortholog assignment. *Kl. lactis* is already included in YGOB. These files were converted to gff format using convert_YGAP_GFF.py and genes were extracted using pull_genes.py. Next, hybrid genomes were created in silico by concatenating fasta sequences of the genes of each species. mRNA and gDNA reads were aligned to the hybrid genomes on a computer cluster using the script aln_reads.py, which calls Bowtie 2 (*Langmead and Salzberg, 2012*). Default parameters were used, which allow mismatches in Bowtie 2. However, reads that mapped equally well to multiple locations in the genome were removed after alignment using discard_multimapping.py. Because ribosomal proteins are highly conserved, they contain stretches of more than 50 bp that are identical between the orthologs belonging to each species in the hybrid. Thus, this filtering step is necessary to assure reads map uniquely to the ortholog from one species or the other.

To quantify differential allelic expression, the reads aligning to each gene were counted using ASE_server.py. mRNA counts of the three *Kl. lactis* × *Kl. marxianus* replicates and seven of the nine *Kl. lactis* × *Kl. wickerhamii* hybrids that were sequenced were highly similar. The two other isolates showed that one of the genomes in the hybrid was present at a lower level than the other suggesting it had been lost from some of the cells (although each of the two isolates lost a different parental genome). The seven reproducible isolates were then treated as replicates for the rest of the analysis. Genome sequencing also revealed that there were two copies of the *Kl. marxianus* genome in each of the *Kl. lactis* × *Kl. marxianus* hybrids, suggesting that our parental *Kl. marxianus* strain was a diploid. The mRNA read counts for each gene in each replicate were then normalized to the total reads in the experiment. Second, they were divided by the gDNA read counts from each gene, thus controlling for the effect of two *Kl. marxianus* genomes in the *Kl. lactis* × *Kl. marxianus* hybrids. (The gDNA read counts per gene were averaged across replicates for each hybrid, so all the replicates were divided by the same gDNA count). Finally, the *Kl. lactis* ortholog read count was divided by either the *Kl. marxianus* or the *Kl. wickerhamii* ortholog to get a differential allelic expression value for each ortholog pair in each replicate.

To calculate significance of differential allelic expression, a two-sided one-sample t-test was used on the log2(differential allelic expression), across replicates (three replicates in the case of the *Kl. lactis* × *Kl. marxianus* hybrid and seven in the case of the *Kl. lactis* × *Kl. wickerhamii* hybrid). The significance of each gene was calculated using the Benjamini-Hochberg procedure to control the false-discovery rate at 0.05. To test for concerted differential allelic expression in the ribosomal proteins, as well as across all gene ontology (GO) terms, the geometric mean of each group of genes was calculated, then tested by the hypergeometric test to see if they were enriched for genes at least 1.1-fold up or down in *Kl. lactis*. Altering this fold cutoff had little effect on the results. These tests were carried out using the ASE_local.py script.

## Protein expression and purification

His$_6$-Rap1 (*S. cerevisaie*) used in main text gel shift experiments was expressed using a previously generated pET28a-Rap1 expression vector in Rosetta II DE3 *E. coli* and purified as described (*Johnson and Weil, 2017*). In brief, after inducing expression in *E. coli* cells grown to an OD$_{600}$ of 0.5 – 1 for 4 hr with 1 mM IPTG at 37°C, cell pellets from 500 mL of culture were resuspended in 20 ml of Rap1 Lysis/Wash buffer (25 mM HEPES-NaOH (pH 7.6), 10% v/v glycerol, 300 mM NaCl, 0.01% v/v Nonidet P-40, 1 mM Benzamidine, 0.2 mM PMSF) and lysed via treatment with 1 mg/mL lysozyme and sonication. Lysate cleared via centrifugation was incubated with 2.5 mL Ni-NTA agarose

(Qiagen) equilibrated with Rap1 Lysis/Wash buffer for 3 hr at 4°C to allow for $His_6$-Rap1 protein binding. Following 3 washes with Rap1 Lysis/Wash buffer, Ni-NTA agarose-bound proteins were transferred to a disposable column and eluted using Rap1 Lysis/Wash buffer containing 200 mM Imidazole.

To generate *S. cerevisiae* Mcm1 for the gel shift and Far Western protein-protein binding assays presented in the main text, *S. cerevisiae MCM1* was cloned into a vector that would allow its expression and purification with an N-terminal MBP tag and PreScission protease cleavage site. Specifically, *S. cerevisiae MCM1* was amplified from *S. cerevisiae* genomic DNA and cloned into a p425 GAL1 expression vector (*Mumberg et al., 1994*) containing an MBP-3C tag (*Feigerle and Weil, 2016*) using the SpeI and XhoI restriction enzymes. MBP-3C-Mcm1 vector was expressed in yeast grown to an $OD_{600}$ of ~3 in 1% w/v raffinose via induction with 2% w/v galactose for 3 hr at 30°C. Yeast cell pellets obtained from 1L of culture were resuspended in 4 mL Mcm1 Lysis Buffer (20 mM HEPES-KOH (pH 7.6), 500 mM potassium acetate, 10% v/v glycerol, 0.5% v/v Nonidet P-40, 1 mM DTT +1X protease inhibitors (0.1 mM PMSF, 1 mM Benzamidine HCl, 2.5 µg/mL aprotinin, 2.5 µg/mL leupeptin, 1 µg/mL pepstatin A)). Cells were lysed via glass bead lysis. Soluble cell extract was obtained via centrifugation and mixed with 2 mL DE-52 resin pre-equilibrated with Mcm1 Lysis Buffer for 5 min at 4°C. Flowthrough from the DE-52 purification was collected and diluted with 20 mM HEPES-KOH, 10% v/v glycerol, and protease inhibitors to reduce the concentration of potassium acetate and Nonidet P-40 to 200 mM and 0.2% v/v, respectively. Binding to 600 µl amylose resin was performed in batch for 2 hr at 4°C. Amylose resin-bound proteins were transferred to a disposable column, washed with 10 column volumes of Mcm1 Wash Buffer (20 mM HEPES-KOH, 200 mM potassium acetate, 10% v/v glycerol, and protease inhibitors) and eluted with 10 column buffers of Mcm1 Wash Buffer containing 10 mM maltose. For gel shift reactions, the N-terminal MBP tag was removed by incubating multiple 50 µl reactions each containing 12 pmol MBP-Mcm1 and 48 pmol lab-generated 3C protease for 30 min at 4°C.

Taf1-TAP TFIID used for Far Western protein-protein binding analyses was purified via a modified tandem affinity protocol as previously described (*Feigerle and Weil, 2016*). GST-, $His_6$-Taf3, $His_6$-Taf4 and GST-Taf4 were expressed in *E. coli* and purified via chromatographic methods that varied upon each protein (*Layer et al., 2010*).

To generate material for the gel shift experiments in the supplement, the full-length Rap1-$His_6$ protein from *Kl. lactis* was purified from *E. coli*. Rap1 was amplified from genomic DNA and cloned into the pLIC-H3 expression vector using XmaI and XhoI to make pTS207. This protein was purified using a $His_6$ tag as previously described (*Lohse et al., 2013*). The full-length Mcm1-HA protein from *Kl. lactis* and *S. cerevisiae* was purified from *S. cerevisiae*. The genes were amplified from genomic DNA and cloned into p426 Gal1P-MCS (ATCC 87331) using BamHI and HindIII to make pTS226 (*S. cerevisiae* Mcm1) and pTS227 (*Kl. lactis* Mcm1). These plasmids were transformed into *S. cerevisiae* W303 for expression. Cells were grown 12 hr in SC –Leu at 30 °C, then in YPGL (1% yeast extract, 2% peptone, 3% glycerol, 2% lactate) for 10 – 12 hr. Then cells were diluted into 1L YPGL to $OD_{600}$ = 0.3 and grown for 4 – 5 hr until $OD_{600}$ = 0.6. Cells were induced with 2% galactose (added from a 40% galactose stock) for 1 hr. Then cells were pelleted, resuspended in an equal volume of lysis buffer, and pipetted into liquid nitrogen to make pellets. Cells were lysed in a Cryomill (Retsch) 6 times for 3 min at 30 Hz, refreezing in liquid nitrogen between cycles.

Cell powder was thawed on ice and diluted to 4 mL/g frozen pellet with lysis buffer (100 mM Tris pH 8.0, 1 mM EDTA, 10 mM fresh β-mercaptoethanol, 10% glycerol, 200 mM NaCl). Lysate was resuspended by pipetting, then cleared by centrifugation for 2 hr at 200,000 x g. Protein was bound to 250 µl of HA-7-agarose (Sigma) slurry per liter yeast culture for 1 hr at 4 °C. Lysate was applied to a Polyprep mini disposable gravity column (Biorad) and washed 4 times with 10 mL lysis buffer. The protein was eluted 4 times with one bed volume of 1 mg/mL HA peptide in lysis buffer after incubating 30 min on a tilt board at room temperature. The protein was stored in aliquots at −80 °C.

Extract for gel shifts was prepared as previously described (*Baker et al., 2011*). 25 mL cultures of cells were grown to OD = 0.8 – 1.0, and frozen at −80 °C. Cells were lysed in 300 µl extract gel shift buffer (100 mM Tris pH 8.0, 200 mM NaCl, 1 mM EDTA, 10 mM $MgCl_2$, 10 mM β-mercaptoethanol, 20% glycerol, EDTA-free protease inhibitor cocktail (Roche)) with 200 µl glass beads on a vortexer for 30 min. Lysate was cleared by centrifugation at 18,000 x g for 20 min and diluted for experiments.

## Gel-mobility shift assays

For the main text experiments, purified *S. cerevisiae* His$_6$-Rap1 and Mcm1 were incubated either individually or in combination in increasing amounts as indicated in the figure legend. All binding reactions were performed using 10 fmol (7000 cpm) of a 79 bp $^{32}$P-labeled fragment of the *Kl. lactis RPS23* promoter containing the Rap1 and Mcm1 binding sites generated via PCR, EcoRI restriction enzyme digestion, and native PAGE purification. Binding reactions were performed in binding buffer (20 mM HEPES-KOH (pH 7.6), 10% v/v glycerol, 100 mM KCl, 0.1 mM EDTA, 1 mM DTT, 25 µg/µl BSA, 25 µg/µl Poly(dG-dC) (double-stranded, alternating copolymer) in a final volume of 20 µl. For competition reactions, binding was performed in the presence of 100-fold molar excess of cold Rap1 WT (R$_{WT}$) or Rap1 scrambled (R$_{sc}$) sequences and/or the Mcm1 WT (M$_{WT}$) or Mcm1 scrambled (M$_{sc}$). Reactions were allowed to proceed for 20 min at room temperature before loading onto 0.5X TBE-buffered (44.5 mM Tris, 44.5 mM Boric acid, 1 mM EDTA (pH 8.0)) 5% polyacrylamide gels and electrophoresed for 1 hr at 200V at room temperature. Gels were vacuum dried and $^{32}$P-DNA signals detected via K-screen imaging using a BioRad Pharos FX imager.

Gel shift experiments in the supplemental material were performed as previously described (*Lohse et al., 2010*). Binding reactions were carried out in 20 mM Tris pH 8.0 50 mM potassium acetate, 5% glycerol, 5 mM MgCl$_2$, 1 mM DTT, 0.5 µg/µl BSA, 25 µg/µl poly(dI-dC) (Sigma).

## Far western assay

Purified proteins tested for direct interaction with Mcm1 in Far western protein-protein binding assays were separated on parallel 4 – 12% NuPAGE Bis-Tris polyacrylamide gels (Life Technologies). 0.5 pmol Taf1-TAP TFIID, 1 pmol His$_6$-Taf3, and 1 pmol His$_6$-Taf4 were used in the assay to test for direct Mcm1 interaction with TFIID subunits. In the assay used to identify the Taf4 Mcm1 binding domain,~0.4 pmol of each Taf4 form, 0.8 pmol His$_6$-Taf3, and 0.8 pmol GST were used. For each experiment, one gel was stained using Sypro Ruby protein stain (Invitrogen) to monitor protein integrity and amount. The other gel(s) were transferred to PVDF membranes pre-equilibrated in transfer buffer. Following transfer, PVDF membranes were incubated with renaturation buffer (20 mM HEPES-KOH pH 7.6, 75 mM KCl, 25 mM MgCl$_2$, 0.25 mM EDTA, 0.05 v/v% Triton X-100 and 1 mM DTT freshly added) for 90 min at 4°C on a tiltboard. The PVDF membrane was then blocked using 5% non-fat milk in renaturation buffer for 30 min at room temperature on a tiltboard. The overlays were performed overnight with 10 nM MBP or 10 nM MBP-Mcm1 with 1% BSA (Roche) as a nonspecific competitor in renaturation buffer. Bound MBP- or MBP-Mcm1 was detected using a standard immunoblotting protocol (primary antibody MBP (NEB Catalog#E8032), used at a dilution of 1:5,000, secondary antibody horse anti-mouse IgG, HRP-linked (Cell Signalling Catalog#7076), used at a dilution of 1:5,000). Detection of bound proteins was achieved via incubation with ECL (GE) and X-ray film.

## Data and code availability

Interspecies hybrid expression data is available at the Gene Expression Omnibus (GEO) repository under accession number GSE108389. Flow cytometry data is available at Flow Repository under accession numbers FR-FCM-ZYWS, FR-FCM-ZYWT, FR-FCM-ZYWU, FR-FCM-ZYWV, FR-FCM-ZYJZ, FR-FCM-ZYJY, and FR-FCM-ZYJ2. Code used in computational analyses is available at doi.org/10.5281/zenodo.1341284.

## Acknowledgements

We thank K Verba, L Pack, S Liang, C Zhou, K Pollard, and E Chow for experimental and analysis guidance. We thank M Lohse, I Nocedal, S Singh-Babak, V Hanson-Smith and other members of the Johnson lab for technical guidance and comments on the manuscript. We thank S Åstrom for use of the *Kl. dobzhanskii* genome prior to publication. We thank I Grigoriev, J K Magnuson, P Inderbitzin, M Nowrousian, A Grum-Grzhimaylo, K O'Donnell, G Bonito and the Metatranscriptomics of Forest Soil Ecosystems project, D Greenshields, J Crouch, F Martin and the Mycorrhizal Genomics Initiative, J Spatafora, and R Gazis for providing access to unpublished genome data produced by the U.S. Department of Energy Joint Genome Institute, a DOE Office of Science User Facility, supported by the Office of Science of the U.S. Department of Energy under Contract No. DE-AC02-05CH11231.

This research was supported by the grants R01 GM115892 and R01 GM037049. ANJ was supported by T32NS00749 from the National Institutes of Health. TRS was supported by a Graduate Research Fellowship from the National Science Foundation.

## Additional information

### Funding

| Funder | Grant reference number | Author |
|---|---|---|
| National Institutes of Health | GM115892 | Amanda N Johnson<br>Jordan T Feigerle<br>P Anthony Weil |
| National Institutes of Health | GM037049 | Trevor R Sorrells<br>Conor J Howard<br>Candace S Britton<br>Kyle R Fowler<br>Alexander D Johnson |

The funders had no role in study design, data collection and interpretation, or the decision to submit the work for publication.

### Author contributions

Trevor R Sorrells, Conceptualization, Data curation, Software, Formal analysis, Validation, Investigation, Visualization, Methodology, Writing—original draft, Project administration, Writing—review and editing; Amanda N Johnson, Conceptualization, Validation, Investigation, Visualization, Methodology, Writing—original draft, Writing—review and editing; Conor J Howard, Candace S Britton, Kyle R Fowler, Investigation, Writing—review and editing; Jordan T Feigerle, Resources, Investigation, Methodology, Writing—review and editing; P Anthony Weil, Conceptualization, Resources, Supervision, Funding acquisition, Writing—review and editing; Alexander D Johnson, Conceptualization, Resources, Supervision, Funding acquisition, Writing—original draft, Writing—review and editing

### Author ORCIDs

Trevor R Sorrells http://orcid.org/0000-0002-3527-8622
Conor J Howard http://orcid.org/0000-0001-5375-6248

### Decision letter and Author response

Decision letter https://doi.org/10.7554/eLife.37563.040
Author response https://doi.org/10.7554/eLife.37563.041

## Additional files

### Supplementary files

• Supplementary file 1. A list of plasmids created for and used in this study. The columns indicate the plasmid name, intended species for use, and purpose for creating the plasmid. A reference is given for plasmids from previous studies.
DOI: https://doi.org/10.7554/eLife.37563.017

• Supplementary file 2. A list of strains created for and used in this study. The columns indicate the strain name, species, and genotype. A reference is given for strains from previous studies.
DOI: https://doi.org/10.7554/eLife.37563.018

• Supplementary file 3. A list of the genomes used for computational analyses. The table indicates the genus and species, assembly version, and the publication or website from which the genome was obtained.
DOI: https://doi.org/10.7554/eLife.37563.019

• Supplementary file 4. Motifs used in the analysis of *cis*-regulatory sequences. This file contains log odds matrices for each of the motifs used throughout the study. The columns indicate the scores for bases A, C, G, and T, respectively. The source of each motif is indicated in the methods section.
DOI: https://doi.org/10.7554/eLife.37563.020

• Supplementary file 5. Mcm1 sites at the RPGs in each species. This file contains the sequence with the highest score matching the Mcm1 position weight matrix at each RPG in each species. The first column is the species identifier, followed by the gene name, position weight matrix score, location of the match, sequence of the match, and the strand of the match. For the purposes of the computational analysis, an Mcm1 site was considered present if the score was above 6.0.
DOI: https://doi.org/10.7554/eLife.37563.021

• Transparent reporting form
DOI: https://doi.org/10.7554/eLife.37563.022

## Data availability

Interspecies hybrid expression data is available at the Gene Expression Omnibus (GEO) repository under accession number GSE108389. Flow cytometry data is available at Flow Repository under accession numbers FR-FCM-ZYWS, FR-FCM-ZYWT, FR-FCM-ZYWU, FR-FCM-ZYWV, FR-FCM-ZYJZ, FR-FCM-ZYJY, and FR-FCM-ZYJ2. Code used in computational analyses is available at doi.org/10.5281/zenodo.1341284.

The following datasets were generated:

| Author(s) | Year | Dataset title | Dataset URL | Database, license, and accessibility information |
|---|---|---|---|---|
| Trevor R Sorrells | 2018 | Interspecies hybrid expression data | https://www.ncbi.nlm.nih.gov/geo/query/acc.cgi?acc=GSE108389 | Publicly available at the NCBI Gene Expression Omnibus (accession no: GSE108389). |
| Trevor R Sorrells | 2018 | Flow cytometry data for Figure 1C, S2A | https://flowrepository.org/id/FR-FCM-ZYWS | Publicly available at FlowRepository (accession no. FR-FCM-ZYWS) |
| Trevor R Sorrells | 2018 | Flow cytometry data for Figure 3E Rps23 | https://flowrepository.org/id/FR-FCM-ZYWT | Publicly available at FlowRepository (accession no. FR-FCM-ZYWT) |
| Trevor R Sorrells | 2018 | Flow cytometry data for Figure 3E Rps17 | https://flowrepository.org/id/FR-FCM-ZYWU | Publicly available at FlowRepository (accession no. FR-FCM-ZYWU) |
| Trevor R Sorrells | 2018 | Flow cytometry data for Figure 3F | https://flowrepository.org/id/FR-FCM-ZYWV | Publicly available at FlowRepository (accession no. FR-FCM-ZYWV) |
| Trevor R Sorrells | 2018 | Flow cytometry data for Figure 3H | https://flowrepository.org/id/FR-FCM-ZYJZ | Publicly available at FlowRepository (accession no. FR-FCM-ZYJZ) |
| Trevor R Sorrells | 2018 | Flow cytometry data for Figure 6A | https://flowrepository.org/id/FR-FCM-ZYJY | Publicly available at FlowRepository (accession no. FR-FCM-ZYJY) |
| Trevor R Sorrells | 2018 | Flow cytometry data for Figure S2B | https://flowrepository.org/id/FR-FCM-ZYJ2 | Publicly available at FlowRepository (accession no. FR-FCM-ZYJ2) |

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
