## [Decision Letter]

Thank you for submitting your article "Intrinsic cooperativity potentiates parallel cis-regulatory evolution" for consideration by *eLife*. Your article has been reviewed by Patricia Wittkopp as the Senior Editor, a Reviewing Editor, and three reviewers. The following individual involved in review of your submission has agreed to reveal his identity: Brian P.H. Metzger (Reviewer #2).

The reviewers have discussed the reviews with one another and the Reviewing Editor has drafted this decision to help you prepare a revised submission.

Summary:

All three reviewers were overwhelmingly positive and believe that this manuscript is an important and rigorous contribution that presents a novel mechanism by which transcriptional regulatory networks can evolve.

Essential revisions:

The consensus is that no additional experiments or analyses are required. Nonetheless, the authors should revise the manuscript in the following ways:

1) Provide a more specific model(s) about how new Mcm1 binding sites could have been selected for and/or maintained, even if this model(s) should be tested in more detail by future studies.

2) Clearly document the sources of all genome sequences analyzed and, for unpublished data, verify that the present analyses meet any terms of use the authors may have agreed to.

3) Provide sufficient detail about the phylogenetic analyses that the results can be replicated.

*Reviewer #1:*

Several cases have been observed where large regulatory networks have acquired binding sites for the same transcription factor (TF) across dozens of genes during evolution. In previous cases, these seemingly improbable events have been shown to occur through facilitation by transposable elements or protein-protein interactions among TFs. The present work provides an attractive new model where the interactions of two TFs with a third protein (TFIID) seem to have facilitated the acquisition of binding sites.

In general, the model was well articulated, and the experiments performed were insightful and rigorous. The diverse methods included hybridization, gene expression analyses, protein biochemistry, genetic reporter tests of cooperativity, and a particularly insightful test of the importance of the second TF in allowing weak binding sites from the first TF to have a stronger effect (Figure 6A).

Nonetheless several major issues should be addressed:

The phylogenetic and ancestral state reconstruction analyses are not sufficiently documented, do not appear to be up to modern standards, and do not discuss conflicts with published phylogenomic analyses that are likely more rigorous.

Subsection “Gains of functional Mcm1 cis-regulatory sites”, "data not shown" does not meet *eLife* reporting standards for any type of data or analyses.

The trend of gene expression for *K. lactis* versus the other species is in the opposite direction from what one might have expected under a simple model where binding sites were adaptively added to increase expression. The authors propose a compensatory model in subsection “Gains of functional Mcm1 cis-regulatory sites”, but they do not explore or discuss much how this would work, which somewhat undermines an otherwise neat and tidy story.

Were promoters analyzed separately that had TATA boxes versus those that were TATA-less? How would this impact conclusions?

Can the authors clarify that JGI is specifically allowing their unpublished genomes to be analyzed for this purpose? Downloading genomes from MycoCosm requires users to agree to terms of use. The list of investigators acknowledged suggests the authors have sought the necessary permissions, but it would be best to be certain, especially since JGI often considers phylogenomic analyses to be reserved for the publication of the genome.

No table of genome versions and citations is provided.

*Reviewer #2:*

The paper by Sorrells et al. addresses how binding sites for the same transcription factor can be gained across dozens of genes in independent lineages. The paper hypothesizes and tests a novel mechanism that the shared binding of two TFs with mediators of transcription can result in a bias of mutations towards functional effects on expression. Overall the paper is excellent, using multiple lines of evidence from genomics through functional tests to suggest that this mechanism may be important not only for the specific example studied but also more generally. I have only a few concerns that I believe are easily addressed.

1) A false dichotomy is set up in subsection “Gains of functional Mcm1 cis-regulatory sites”. It may be that Mcm1 doesn't change the overall expression, but instead changes regulation. This is addressed slightly in the next session, but the given impression is too clean given the lack of data that directly addresses this question.

2) Regardless of this concern, I like the experimental design. However, the species aren't that different in Mcm1 binding gains, i.e. both are in the full gain group. An alternative explanation for the results is that the organisms chosen simply aren't that different in their RPG regulation compared to what would happen upon the initial gain of Mcm1 binding sites. I believe that this needs to be acknowledged.

3) Compensation seems to imply an order, i.e. that the Mcm1 gain was after the loss of some other TF binding site. It seems more likely it was the other way around, that an Mcm1 gain permitted the loss of other binding sites. However, the extent to which the addition of Mcm1 binding site alone directly influenced expression isn't tested (i.e. adding a Mcm1 binding site to a promoter from a species without the Mcm1 gain). If the simple addition of Mcm1 alters expression, it becomes unclear how the authors think evolution proceeded given the strong purifying selection on RPG expression.

4) The figures don't show that Rap1^AS7Ala^ doesn't act cooperatively as stated.

5) The experiments only address the mediator binding hypothesis from the Rap1 perspective, not Mcm1 (which is the thing that is new), nor TFIID. While these experiments aren't necessary, I think this omission needs to be acknowledged.

*Reviewer #3:*

This is a convincing and enjoyable manuscript that presents evidence that multiple independent co-options of the transcription factor Mcm1 to regulate ribosomal protein genes (RPGs) occurred in different yeast clades (about 7-9 independent occurrences), building on the authors' previous work in Tuch et al. (2008a). The proposed mechanism is a pre-existing ability of Mcm1 to cooperate with Rap1, through their separate interactions with TFIID. The authors' evolutionary model is convincing and provides an explanation for several otherwise puzzling aspects such as the recurrence of particular spacings between Mcm1 and Rap1 sites. As always with Johnson lab papers, this one is very well written, the figures are excellent, and the experiments are rigorous. The work is of broad significance, because it provides a general rationale for how a cis-regulatory motif can abruptly emerge in a large set of genes such as the RPGs.

The one question that the manuscript did not seem to answer adequately is what is the selective pressure to maintain Mcm1 sites in RPG genes after they arise by chance (model in Figure 7)? Since these sites do not contribute to increased transcription of RPGs, I cannot see what prevents them from decaying again. The text mentions this issue: the Discussion section says that new Mcm1 sites at RPGs "would fall under selection as other cis-regulatory sites deteriorated by mutation" (citing Figure 7, though Figure 7 doesn't illustrate this situation) but leaves me wondering what TFs bind to these other sites, or if there is evidence to support this idea. In principle, if the idea is correct, the deterioration of the other sites should be evident in Figure 1—figure supplement 1. For example, Sfp1 sites seem to disappear in the *Kluyveromyces* clade around the same time as Mcm1 sites appear, so is Sfp1 a candidate?

---

## [Author Response]

Summary:All three reviewers were overwhelmingly positive and believe that this manuscript is an important and rigorous contribution that presents a novel mechanism by which transcriptional regulatory networks can evolve.Essential revisions:The consensus is that no additional experiments or analyses are required. Nonetheless, the authors should revise the manuscript in the following ways:1) Provide a more specific model(s) about how new Mcm1 binding sites could have been selected for and/or maintained, even if this model(s) should be tested in more detail by future studies.

A model for how Mcm1 binding sites could be selected for requires two things: that the Mcm1 site is functional and that there is an evolutionary force that tends to preserve the site. We have focused primarily on a mechanism (intrinsic cooperativity) that causes weak Mcm1 sites to be functional in RPGs as they arise by mutation.

We can think of two plausible mechanisms explaining the preservation of Mcm1 sites once they arise. One mechanism would be direct selection for the increase of expression of the RPG, or the RPGs as a whole. This is plausible because the selection that acts on RPG expression as a whole changes significantly between species (as evidenced by our interspecies hybrid experiment as well as others). The selection acting on individual RPGs also probably changes on short timescales, potentially resulting on selection for a subset RPGs to have increased expression levels. However, we have no direct evidence for selection for increased expression of RPGs leading to the gain of Mcm1 sites.

We have direct evidence for a second model: a compensatory model in which the Mcm1 sites counteract losses of other regulatory elements. This is supported by the hybrid experiment results and cis-regulatory site analysis. The compensatory model is not incompatible with the first model (for example if selection for increased expression of RPGs alternated with relaxed selection or selection to reduce expression of RPGs). However, it could also be effected by neutral forces: if Mcm1 sites are allowed to be weak, they could present a smaller “mutational target” than the strong sites for another transcription regulator, resulting over time in the increase in Mcm1 sites at the expense of the other regulator due to mutational processes.

Compensatory changes could plausibly occur in multiple different orders on the scale of individual mutations, cis-regulatory sites, and genes. For example, mutations affecting the cis-regulatory sites could be fixed simultaneously (on the same haplotype) or alternate between strengthening the Mcm1 sites and weakening a second site. Alternatively, the Mcm1 site could become fixed some period of time before the loss of a second site (or vice-versa). We suspect that the precise order may differ for each gene as the Mcm1 sites were gained over millions of years.

We have done our best to indicate these scenarios in the text of the manuscript while making it clear what we have direct evidence for. We have modified Figure 7 to indicate the types of selection that could maintain Mcm1 sites, but because there are multiple plausible scenarios we have not directly illustrated them.

2) Clearly document the sources of all genome sequences analyzed and, for unpublished data, verify that the present analyses meet any terms of use the authors may have agreed to.

We have added Supplementary file 3 which includes the sources and versions of the genome assemblies and annotations used. The principle investigators of all of the unpublished genomes used in this study were contacted and shown an example analysis (or the final analysis) and consented to the use of the genomic data for the purpose of this publication. This consent was obtained prior to the preprint and initial submission of the manuscript. In all, 24 genomes of interest were excluded from the analysis because investigators did not consent to their use. Consent from a subset of the investigators was confirmed during revision and all gave the same response.

3) Provide sufficient detail about the phylogenetic analyses that the results can be replicated.

Additional detail was added to the methods to facilitate replication of the phylogenetic analyses. Specifically, we have added details about the tree construction, scoring of intergenic regions for cis-regulatory sites, and the newly added stochastic character mapping analysis to more rigorously estimate gains and losses. In addition to the scripts included in the initial submission, we have included Supplementary file 3 with the genome versions and Supplementary file 4 with the position weight matrices used for identifying the cis-regulatory sequences. We have also included a list of the ribosomal protein genes in each species and the presence of Mcm1 sites in Supplementary file 5.

Reviewer #1:[…]The phylogenetic and ancestral state reconstruction analyses are not sufficiently documented, do not appear to be up to modern standards, and do not discuss conflicts with published phylogenomic analyses that are likely more rigorous.

We have provided a reference and additional details in the methods on how the phylogenetic tree was created. Because we focus on the mechanism of parallel evolution and transcription activation, a detailed discussion of the tree topology is beyond the scope of the present study. Several differences in topology when compared to published studies may be ascribed to the use of unpublished genomes, but discussion of this is reserved for the principal investigators of these genome sequencing projects.

In response to this comment we have also implemented a new estimation of gains and losses to better quantify the uncertainty in this analysis. The model for evolution of binding sites is one of the most challenging questions we faced. In the end, rather than choosing a model specific to cis-regulatory sites whose assumptions were violated by the data we had, we chose a simple model of character evolution. (To be specific, the only sequence-independent model of binding site evolution we are aware of is CRETO from Otto et al. (2009) and to give an accurate estimate of evolutionary rates, it requires that the half-life of binding sites is on the order of the length of the tree from root to tip, whereas in our data Mcm1 site turnover is happening many times from root to tip.) Because of the large evolutionary distance, we decided to categorize species into presence or absence of Mcm1 sites and treat this as a discrete character under two different models. We strongly encourage the development of better computational models for cis-regulatory site evolution by experts in that area.

Subsection “Gains of functional Mcm1 cis-regulatory sites”, "data not shown" does not meet eLife reporting standards for any type of data or analyses.

We have further explained these results in the text of the manuscript and removed the assertion about phylogenetic branches because we did not rerun the analysis under multiple different tree topologies.

The trend of gene expression for K. lactis versus the other species is in the opposite direction from what one might have expected under a simple model where binding sites were adaptively added to increase expression. The authors propose a compensatory model in subsection “Gains of functional Mcm1 cis-regulatory sites”, but they do not explore or discuss much how this would work, which somewhat undermines an otherwise neat and tidy story.

Our intention was to present the results without trying to force them into a neat and tidy story. However, as suggested by all reviewers, we provide additional details about how a compensatory model may work, the details of which would need to be tested in future studies. Briefly, the scale of evolution between the interspecies hybrid may be much longer than selection for increased expression of the genes, or the model could involve neutral mutational processes alone.

Were promoters analyzed separately that had TATA boxes versus those that were TATA-less? How would this impact conclusions?

We analyzed whether Rap1 or Mcm1 sites showed a relationship with the strength or location of TBP/Spt15 binding sites, but no clear pattern was observed. In general, yeast ribosomal protein genes lack strong TATA boxes which may explain their dependence on transcription regulators such as Rap1 and Mcm1 that can directly interact with TFIID.

Can the authors clarify that JGI is specifically allowing their unpublished genomes to be analyzed for this purpose? Downloading genomes from MycoCosm requires users to agree to terms of use. The list of investigators acknowledged suggests the authors have sought the necessary permissions, but it would be best to be certain, especially since JGI often considers phylogenomic analyses to be reserved for the publication of the genome.No table of genome versions and citations is provided.

See our response to the essential revisions.

Reviewer #2:[…]1) A false dichotomy is set up in subsection “Gains of functional Mcm1 cis-regulatory sites”. It may be that Mcm1 doesn't change the overall expression, but instead changes regulation. This is addressed slightly in the next session, but the given impression is too clean given the lack of data that directly addresses this question.

We agree that this seems simplistic because we do not discuss broad vs. condition-specific expression of Mcm1 until a later section. We have modified the text to specify that these hypotheses apply only to the function of Mcm1 during the rapid growth conditions used for the experiments in Figure 1 and Figure 2.

2) Regardless of this concern, I like the experimental design. However, the species aren't that different in Mcm1 binding gains, i.e. both are in the full gain group. An alternative explanation for the results is that the organisms chosen simply aren't that different in their RPG regulation compared to what would happen upon the initial gain of Mcm1 binding sites. I believe that this needs to be acknowledged.

We were indeed disappointed that we were unable to recover hybrids between species with larger differences in the number of Mcm1 binding sites (attempts are described in the methods). Unfortunately, we aren’t aware of any intermating species with a difference greater than that between *Kl. lactis* and *Kl. wickerhamii* (23% more RPGs have Mcm1 sites in *Kl. lactis*). Reports of distantly related species mating are probably due to misidentification of the species before molecular techniques existed.

We have modified the text to indicate our experiments address the question of evolution over the timescale of the divergence between each pair of species. This is a scale in which a significant fraction of RPGs can gain Mcm1 binding sites, suggesting that the gains aren’t completed by this time. One approach to capture the “initial gain” of Mcm1 sites as you suggest would be to measure allele-specific expression in hybrids between species such as *S. cerevisiae* and *S. mikatae* which show a slight difference between the levels of Mcm1 sites at the RPGs. However, we feel this is less informative because there is no guarantee that *S. mikatae* will go on to gain Mcm1 sites at a large fraction of its RPGs as we know happened in *Kluyveromyces*.

3) Compensation seems to imply an order, i.e. that the Mcm1 gain was after the loss of some other TF binding site. It seems more likely it was the other way around, that an Mcm1 gain permitted the loss of other binding sites. However, the extent to which the addition of Mcm1 binding site alone directly influenced expression isn't tested (i.e. adding a Mcm1 binding site to a promoter from a species without the Mcm1 gain). If the simple addition of Mcm1 alters expression, it becomes unclear how the authors think evolution proceeded given the strong purifying selection on RPG expression.

The best way to do this experiment would be to perform ancestral reconstruction on all intergenic sequences to see the effect of each gain of Mcm1 binding site along with other changes that happened in close evolutionary proximity, but this is unfortunately not possible because of the fast rate of evolution of intergenic regions. We suspect however, that the answer of order may differ between each gene depending on where the actual expression of the gene falls in the range of acceptable expression levels. See also our other responses regarding more details about the compensatory model.

4) The figures don't show that Rap1^AS7Ala^ doesn't act cooperatively as stated.

Thank you for pointing this out—we have changed the conclusion to “these mutations strongly reduced 3-AT resistant growth, indicating that efficient expression requires an intact Rap1 activation domain.”

5) The experiments only address the mediator binding hypothesis from the Rap1 perspective, not Mcm1 (which is the thing that is new), nor TFIID. While these experiments aren't necessary, I think this omission needs to be acknowledged.

The experiments that identified the point mutations disrupting the Rap1-TFIID interaction comprised an entire publication (Johnson and Weil, 2017), so doing so with Mcm1 and TFIID is beyond the scope of our current study. Because the intrinsic cooperativity model requires interactions between TFIID and both Rap1 and Mcm1, we feel that disrupting either interaction is a sufficient test of the model.

Reviewer #3:This is a convincing and enjoyable manuscript that presents evidence that multiple independent co-options of the transcription factor Mcm1 to regulate ribosomal protein genes (RPGs) occurred in different yeast clades (about 7-9 independent occurrences), building on the authors' previous work in Tuch et al. (2008a). The proposed mechanism is a pre-existing ability of Mcm1 to cooperate with Rap1, through their separate interactions with TFIID. The authors' evolutionary model is convincing and provides an explanation for several otherwise puzzling aspects such as the recurrence of particular spacings between Mcm1 and Rap1 sites. As always with Johnson lab papers, this one is very well written, the figures are excellent, and the experiments are rigorous. The work is of broad significance, because it provides a general rationale for how a cis-regulatory motif can abruptly emerge in a large set of genes such as the RPGs.The one question that the manuscript did not seem to answer adequately is what is the selective pressure to maintain Mcm1 sites in RPG genes after they arise by chance (model in Figure 7)? Since these sites do not contribute to increased transcription of RPGs, I cannot see what prevents them from decaying again. The text mentions this issue: the Discussion section says that new Mcm1 sites at RPGs "would fall under selection as other cis-regulatory sites deteriorated by mutation" (citing Figure 7, though Figure 7 doesn't illustrate this situation) but leaves me wondering what TFs bind to these other sites, or if there is evidence to support this idea. In principle, if the idea is correct, the deterioration of the other sites should be evident in Figure 1—figure supplement 1. For example, Sfp1 sites seem to disappear in the Kluyveromyces clade around the same time as Mcm1 sites appear, so is Sfp1 a candidate?

Your hypothesis is exactly what we think is happening—in some clades where Mcm1 sites are gained you can find some other regulator site that seems to be at lower levels in that clade (e.g. Fhl1, Sfp1, and Rrn7 in *Kluyveromyces*, Cbf1 in *Pachysolen tannophilus*, Tbf1 in *Yarrowia lipolytica*, Hmo1 in *Hanseniaspora uvarum*). However, there are also species and clades where there is no clear loss associated with Mcm1 site gains. A study from *Saccharomyces* found the differences in overall nucleotide composition of the promoter contributes greatly to expression divergence of RPG expression (Zeevi, 2014) so this may explain these additional species. We have now described this idea more explicitly several places in the manuscript.